# Intracontinental deformation of the Tianshan Orogen in response to India-Asia collision

Wei Li[1,2,3], Yun Chen [1,2 ✉], Xiaohui Yuan [3], Wenjiao Xiao [1,4,5 ✉] & Brian F. Windley[6]

How the continental lithosphere deforms far away from plate boundaries has been long debated. The Tianshan is a type-example of ongoing lithospheric deformation in an intra-continental setting. It formed during the Paleozoic accretion of the Altaids and was rejuvenated in the Cenozoic, which might be a far-field response to the India-Asia collision. Here we present seismic images of the lithosphere across the central Tianshan, which were constructed from receiver functions and Rayleigh wave dispersions along a N–S-trending linear seismic array. We observe an extensively deformed lithosphere in the Tianshan with inherited, structurally controlled brittle deformation in the shallow crust and plastic deformation near the Moho. We find that earlier multiple accretionary structures were preserved in the crust, which was deformed by pure-shear shortening in the south and thick-skinned tectonics in the north but was limitedly underthrusted by surrounding blocks. A balanced cross-section of Moho discontinuities supports the concept that intracontinental deformation in the Tianshan intensified synchronously with the direct contact between the underthrusting Indian slab and the Tarim Craton in the Late Miocene (~10 Ma). These findings provide a robust and unified seismic model for the Tianshan Orogen, and confirm that effective delivery of the India-Asia collision stress induced the rejuvenation of this intracontinental orogen.

[1] State Key Laboratory of Lithospheric Evolution, Institute of Geology and Geophysics, Chinese Academy of Sciences, Beijing 100029, China. [2] CAS Center for Excellence in Deep Earth Science, Guangzhou 510640, China. [3] Deutsches GeoForschungsZentrum GFZ, Potsdam 14473, Germany. [4] Xinjiang Research Center for Mineral Resources, Xinjiang Institute of Ecology and Geography, Chinese Academy of Sciences, Urumqi 830011, China. [5] College of Earth and Planetary Sciences, University of Chinese Academy of Sciences, Beijing 100049, China. [6] School of Geography, Geology and the Environment, University of Leicester, Leicester LE1 7RH, UK. ✉email: yunchen@mail.iggcas.ac.cn; wj-xiao@mail.iggcas.ac.cn

Orogens, viewed as a result of rigid plate collision, are largely confined to plate boundaries[1], whereas orogens inside continental plates are referred to as intracontinental orogens[2]. The Tianshan Orogen (Fig. 1), situated in the southern Altaids[3,4], records long-lived accretion and continental growth in the Paleozoic[3–5]. In the Cenozoic, this orogen was rejuvenated and re-deformed by the India-Asia collision that was thousands of kilometers away[6–8]. Hence, the Tianshan Orogen is an ideal site to investigate ongoing intracontinental deformation.

Numerical models have suggested that rheological heterogeneity of the lithosphere controlled the intracontinental deformation of the Tianshan[9–11], but it remains debatable how the deformation was accommodated across the orogen and what the tectonic trigger was. The Cenozoic deformation of the Tianshan has been related to the progressive growth of the Tibetan Plateau, which created the high gravitational potential energy necessary to transmit the pressure northwards[5,8,9]. Thermochronologic and stratigraphic studies indicate that the intracontinental deformation of the Tianshan commenced at a low strain in the Early Miocene and significantly increased at ~10 Ma[12–14], which was much later than the initial continental collision between India and Asia at ~55 Ma[15,16]. Another model links the Tianshan to the transpressional system of the right-slip shears induced by the oblique convergence between Arabia and Asia[16]. Despite some of the estimated age of continental collision between Arabia and Asia being close to the uplift of the Tianshan[17], it remains unclear whether the soft continental collision suggested there[18] can cause the uplift of the distant Tianshan. Clearly, a better knowledge of

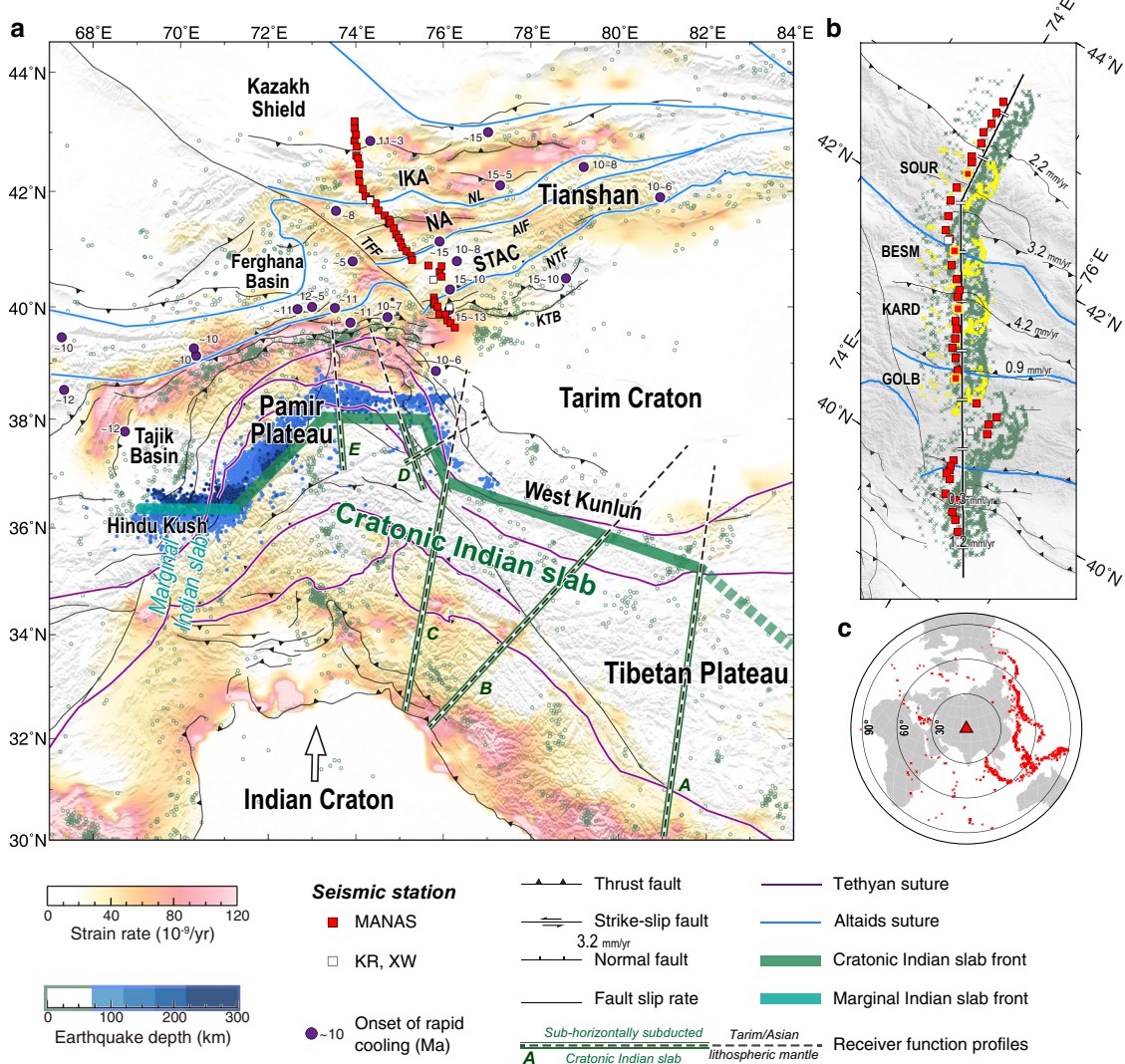

**Fig. 1 Tectonic setting of the Tianshan and data coverage of the seismic cross-section. a** Tectonic map with strain rates inverted from GPS observations[37]. Squares denote seismic stations used in this study from the Middle AsiaN Active Source project (MANAS) and other networks (KR and XW). Green and blue circles show earthquakes at shallow[65] (≤70 km) and intermediate[66,67] (70–300 km) depths. The purple points with numbers denote the onset of rapid cooling derived from thermochronological studies[13,14]. The present-day northern front of the Indian slab[31,49,66], including the marginal Indian slab beneath the Hindu Kush and the cratonic Indian slab beneath the Pamir Plateau and the western Tibetan Plateau are together constrained by the interpretations of the previous receiver function profiles[29–33] (A: Zhao et al.[31], B: Rai et al.[30], C: Kumar et al.[29], D: Xu et al.[33], E: Schneider et al.[32]). Sutures in the Altaids and the Tethyan tectonic domain are marked with light blue and purple lines, respectively, and major faults are in black lines (Central Asia Fault Database[68]). AIF Atbashy-Inylchek Fault, IKA Issyk Kul Arc, NA Naryn Arc, NL Nikolaev Line, NTF North Tarim Fault, KTB Kepingtag Thrust Belt, STAC South Tianshan Accretionary Complex, TFF Talas-Ferghana Fault. **b** Stations and piercing points of the *Ps* phases at a depth of 60 km along the cross-section are indicated by solid lines with 50-km scale marks. Numbers denote fault slip rates[28,69]. Sample stations GOLB, KARD, BESM, and SOUR are highlighted by yellow frames, for which inversions are shown in the supplement. **c** Distribution of events used in this study.

the spatial and temporal development of the structures across the Tianshan could significantly improve the understanding of how and where the continental lithosphere responds to the far-field plate collisions.

Geophysical investigations support the hypothesis that the upper mantle beneath the Tianshan is so weak[19–21] that it accumulated strain from the distant but ongoing India-Asia collision. As a prominent intracontinental deformation orogen, the Tianshan has accommodated ~20-mm/yr N–S shortening, which is approximately half of the present-day India-Asia convergence rate[22,23]. Previous seismic images seem to suggest that the N–S shortening was accommodated by large-scale underthrusting of the Tarim and Kazakh lithospheres[20,24–27]. Had such a scenario existed, most of the N–S shortening would have occurred near the northern and southern discrete boundaries, as is the case in the Himalayas where the Indian slab is underthrusting beneath the Tibetan Plateau (Fig. 1a). However, geodetic and neotectonics observations indicate that the shortening in the Tianshan is distributed throughout the orogen without a significant increase close to the boundaries[22,23,28] (Fig. 1a).

The Cenozoic N–S shortening of the Tianshan shows an along-strike variation and decreases from west to east[22,23]. The central Tianshan, with the western boundary at the Talas-Ferghana Fault, is adjacent to the sub-horizontal underthrusting Indian slab beneath the Pamir Plateau and the West Kunlun[29–33] (Fig. 1a) and experienced the strongest Cenozoic N–S shortening along the entire orogen[23,34]. At the eastern, topographically low, termination of the Tianshan, the horizontal shortening in the Late Cenozoic was only 10–15 km and the deformation changed into the left-lateral transpressional type[8].

Here, we studied the seismic structures of the crust and mantle lithosphere beneath the central Tianshan to provide a comprehensive image of the ongoing intracontinental deformation. The seismic cross-section of this study is close to the western end of the Tianshan Orogen, which is tectonically subdivided, from the north to the south, into the Issyk Kul Arc (IKA), Naryn Arc (NA), and South Tianshan Accretionary Complex (STAC), separated by ophiolitic mélanges and/or large-scale faults[4] (Fig. 1). We collected data mainly from a N–S aligned linear broadband seismic array of the Middle AsiaN Active Source project (MANAS)[25] (Fig. 1), and applied the joint inversion scheme of receiver functions and Rayleigh wave dispersions[35] to construct a S-wave velocity ($V_S$) model, and the common conversion point (CCP) stacking technique[36] to image the geometry of the Moho and intracrustal interfaces. The results exhibit a spatial relationship between the surface deformation and the deep structures of the Tianshan, and further illustrate the mechanism and essential conditions for the intracontinental deformation.

## Results

The $V_S$ model and CCP stacked receiver function image constructed in this study are shown in Fig. 2. The cross-section passes through different tectonic units of the Tianshan Orogen[4], i.e., IKA, NA, and STAC. The $V_S$ model reveals a widespread low-$V_S$ middle-lower crust (~3.4 km/s) beneath the STAC, and a series of elongate low-$V_S$ anomalies (~3.5 km/s) in the middle crust confined to several bands beneath the NA and IKA (Fig. 2d). These bands of low-$V_S$ anomalies correlate well with the negative $Ps$ phases in the CCP image, the southernmost of which can be traced as a clear interface that dips to the south at ~30° beneath the NA (Fig. 2c).

In the CCP image (Fig. 2c), the Moho can be traced by strong positive $Ps$ phases. Along the cross-section, the Moho has been thrust-imbricated into several segments beneath the entire Tianshan Orogen and a doublet beneath the Nikolaev Line (NL).

The deep Moho segments are located at ~70 km depth below the STAC and at ~60 km depth below the IKA, whereas shallow segments are found at 40–50 km depth below the Tarim Craton, NA, and Kazakh Shield. The Moho thrusts appear to correlate with the sub-Moho Vs variations. Contrasting structures in the uppermost mantle are shown from south to north across the NL. A 40-km thick high-$V_S$ mantle lid (~4.5 km/s) lies above the low-$V_S$ anomaly (~4.1 km/s) beneath STAC and NA, whereas a thick high-$V_S$ anomaly (~4.5 km/s) spreads over the entire depth range beneath IKA and Kazakh Shield in the north and beneath Tarim Craton in the south (Fig. 2d).

## Discussion

**Inherited properties control shallow crustal shortening**. Along the profile, a number of active thrust faults can be identified by large slip rates (>2.0 mm/yr)[28] (Fig. 2b) and significantly increased strain rates[37] (Fig. 2a). In our crustal model, we observe several south-dipping low-$V_S$ bands, which extend continuously from the shallow to middle-lower crust (Fig. 2c, d) and correlate well with active faults at the surface. These bands represent crustal shear zones with weak rheology at depth[38]. Given the position and geometry of these crustal structures, most of the shallow crustal shortening in the NA and IKA was likely accommodated by thick-skinned tectonics, which is consistent with the intense seismicity near the shear zones with high-angle thrust focal mechanisms[39]. The high and flat topography of the STAC and the strongly depressed Moho below it are indications of extensive crustal shortening. However, the fault slip rate observed in the STAC is extremely small (~0.3 mm/yr) (Fig. 2b), implying that the STAC underwent a pure-shear shortening with diffusive deformation accommodated by the rheologically weak crust. This model is supported by the stable low stain rate (~$20 \times 10^{-9}$/yr, Fig. 2a) and lack of seismic activity (Fig. 2d).

These differences in shallow crustal deformation structures correlate well with the locations of tectonic terranes accreted in the Paleozoic, i.e., the pure-shear shortening in the STAC contrasts with the thick-skinned tectonics in the NA and IKA (Fig. 2a–d). Magnetotelluric studies also show that the crust of the central Tianshan consists of isolated high resistivity bodies separated by conductive zones in the north, but low resistivities in the southern part adjacent to the Tarim Craton[40]. The STAC was generated by the long-lived consumption of the South Tianshan Ocean in the Paleozoic. It contains high/ultrahigh-pressure rocks, ophiolitic mélanges, and imbricated sedimentary rocks, which are interpreted as off-scraped fragments[4] (Fig. 2b). Such an accretionary wedge was too weak to deliver the stresses and in consequence experienced pure-shear shortening in response to the India-Asia collision, whereas the crust of the Paleozoic volcanic arcs (NA and IKA) to the north was so strong[4,28] that active faults were able to penetrate into the deep crust to accommodate the contractional stress (Fig. 2b–d).

**Moho deformation responses to deep crustal shortening**. Previous models have proposed large-scale underthrusting of the Tarim Craton and/or the Kazakh Shield beneath the Tianshan[24–27]. Our results demonstrate that the underthrusting of the lower crust is only limited in the Tarim Craton and even negligible in the Kazakh Shield, instead, prominent Moho steps are visible near the southern and northern boundaries of the Tianshan that were accompanied by the fade of seismic activities in the lower crust (Fig. 2c, d, see details in the Supplementary Text 1). Another accordance of the underthrusting model is the high-velocity anomalies in the upper mantle of the Tianshan observed in the teleseismic body wave tomographic images[20], which were interpreted as a deeply subducted lithospheric mantle.

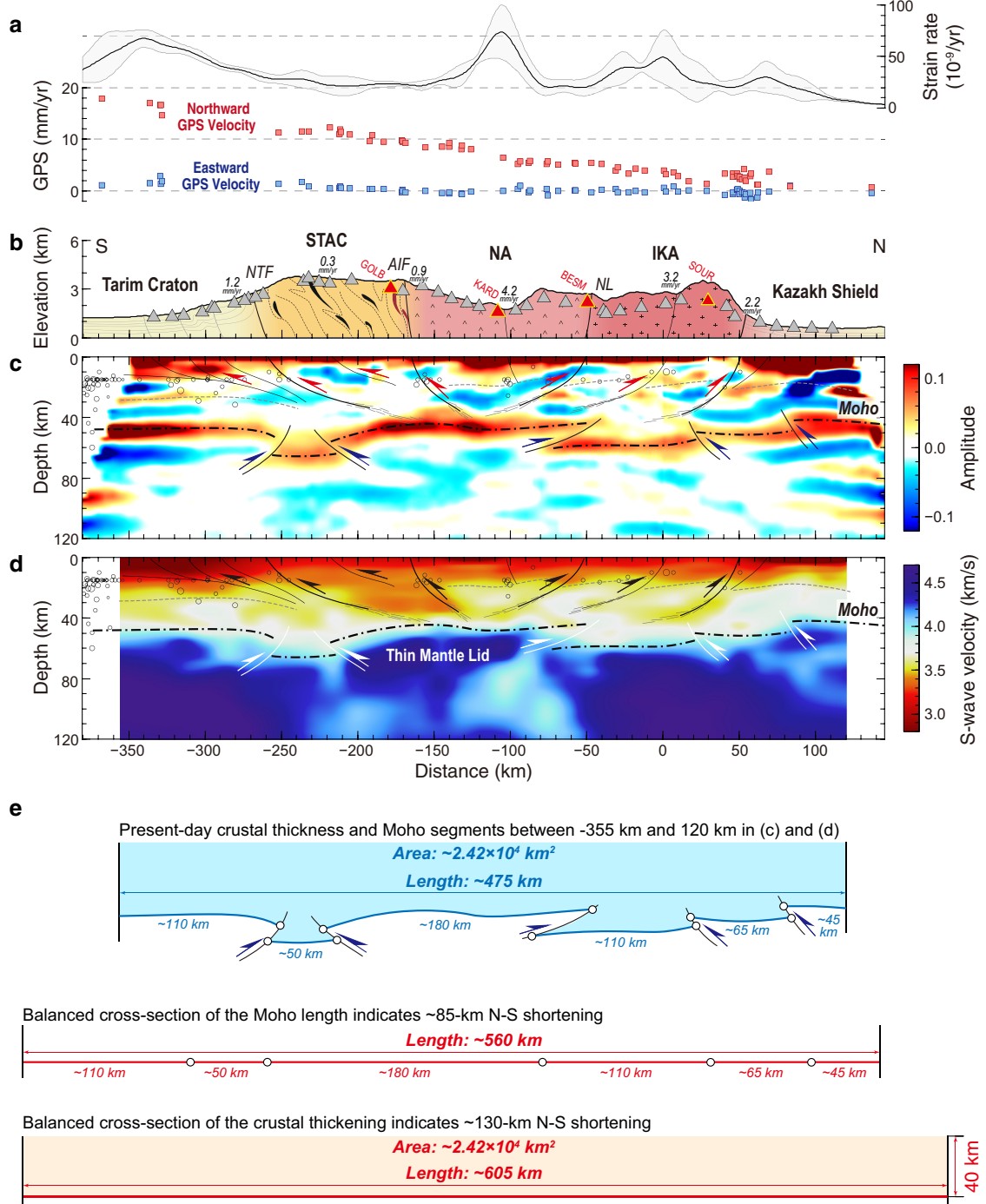

**Fig. 2 Interpreted seismic images and balanced cross-sections across the central Tianshan. a** Strain rates[37] and GPS velocities[70] along the cross-section. The bold black line and gray-filled area indicate the averaged strain rates and corresponding standard deviations within a 1.0-degree width corridor centered by the cross-section. The individual GPS observations within the corridor are projected onto the cross-section as blue squares for eastward components and red squares for northward components. **b** Schematic geological cross-section across the Tianshan showing major tectonic units (modified after Xiao et al.[4]) and the slip rates of main faults[28,69]. Triangles represent stations used in this study, and red ones with marks indicate four stations chosen as examples showing the joint inversion in Supplementary Figs. 4 and 5. AIF Atbashy-Inylchek Fault, IKA Issyk Kul Arc, NA Naryn Arc, NL Nikolaev Line, NTF North Tarim Fault, STAC South Tianshan Accretionary Complex. **c**, **d** The CCP stacking image with a Gaussian coefficient of 2.0 and the V_S model were obtained from joint inversion in this study. These are plotted with interpreted faults (solid lines), Conrad (dashed thin lines) and Moho (dashed thick lines) discontinuities. Gray circles denote earthquakes located within a 50-km width corridor centered by the cross-section[65]. **e** Estimates of N–S shortening in the Tianshan based on the balanced cross-sections of the Moho length and crustal thickening.

However, a large amount of shortening (>500 km, according to the scale of anomalies in Lei & Zhao[20]) expected in the interpreted model is difficult to reconcile with the short rejuvenation history of the Tianshan (~10 Ma)[12–14], even with the current high contractional rate (~20 mm/yr)[22,23].

Based on the almost linear decrease of GPS velocities across the Tianshan, England and Molnar[41] argued that the Tianshan lithosphere has a spatially invariant strength controlled by its low-temperature plasticity, which resulted in plastic deformation around the Moho that has accommodated the stress in the ductile lithosphere. Interestingly, our refined crustal model shows a significant discontinuous Moho with thrust structures across the Tianshan (Fig. 2c, d). Similar Moho discontinuities were imaged in the eastern Tianshan[42,43], although the lateral variation along the strike of the orogen is unclear. It should be noted that the peneplanation in the Tianshan is generally inferred to have developed during the Mesozoic tectonic quiescence[44] after the Paleozoic accretion of the Altaids[4,5]. Accordingly, the Moho can be used as a marker horizon for estimating the amount of N–S shortening across the Tianshan in response to the India-Asia collision in the Cenozoic. As shown in Fig. 2e, we estimate that the Cenozoic N–S shortening was ~85 km by balancing the Moho length and ~130 km by balancing the crustal volume across the Tianshan, respectively. It can be inferred that the significant N–S shortening started at ~11–6 Ma by combining the present-day N–S shortening rates of ~15 mm/yr from GPS observations (Fig. 2a) and ~12 mm/yr from Late Quaternary fault slip rates (Fig. 2b) across the estimated section. Such an inferred time is similar to the onset of rapid uplift of the Tianshan (~10 Ma)[12–14] (Fig. 1a), which confirms that the deep deformation of the central Tianshan is performed as the Moho segments are discontinued by localized thrusts. Therefore, we propose that the plastic deformation close to the Moho mainly accommodated the N–S shortening in the deep lithosphere of the central Tianshan, and that the underthrusting of the Tarim Craton or the Kazakh Shield should be on a limited scale. This deformation model correlates well with the shallow crustal structures revealed in this study and with the comprehensive observations from geodesy and neotectonics, and supports the concept that intracontinental deformation in the Tianshan intensified in the Late Miocene (~10 Ma).

**Rejuvenation of the Tianshan Orogen**. Our images reveal that the lithosphere of the Tianshan was extensively deformed in response to the Cenozoic India-Asia collision. The question remains as why the amount of the India-Asia convergence accommodated in the Tianshan with the N–S shortening of ~20 mm/yr is nearly equivalent to that in the Tibetan Plateau[22,23]. The former should be much lower considering the lithospheric strength of the Tianshan is an order of magnitude greater than that of the Tibetan Plateau[41]. After the break-off of Greater India (i.e., the northern passive continental margin of the Indian plate, as suggested by plate reconstructions and seismic tomography studies[45]), the India-Asia convergence changed from continental subduction to continental collision due to the buoyancy of the Indian Craton with its thick and depleted lithospheric mantle[46,47]. Although the Tibetan Plateau and the Indian Craton are now abutted along the Indus-Yarlung ophiolite belt at the surface, seismic images suggest that the cratonic Indian slab has underthrust sub-horizontally farther northwards for several hundred kilometers beneath the western Tibetan Plateau[31,48]. Previous receiver function studies confirm that the front of the cratonic Indian slab has reached the Tarim Craton in the West Kunlun[29–31] and thrust under the Pamir Plateau northwards to the intermediate-depth seismic zone[32,33] (Fig. 1a). The synchronous foundering of the deep crust beneath the Pamir Plateau at ~11 Ma, documented by the xenoliths in the southeastern Pamir and related to the indentation process of the cratonic Indian slab beneath the Pamir Plateau[49], provides an age record for the initial contact between the Indian slab and the Tarim Craton. The balanced cross-section analysis of the Moho discontinuities in this study (Fig. 2e) and the onset of rapid exhumation in the Tianshan[12–14] jointly support the concept that the intracontinental deformation in the Tianshan intensified after the Late Miocene (~10 Ma) synchronously with the India-Tarim collision. Therefore, we propose that the direct contact between the cratonic Indian slab and the Tarim Craton enabled the compressional stresses to reach the distant intracontinental setting[50] and intensified the rejuvenation of the Tianshan Orogen. Such a stress transformation in the upper mantle was a significant contributor to the slowing-down of the India-Asia convergence[51] and to the clockwise rotation of the Tarim Craton[34], but also can well explain why the Tianshan was rejuvenated much later than the initial India-Asia collision[11].

Previous studies have also reported low-velocity anomalies in the upper mantle beneath the Tianshan[19–21], implying a high temperature and weak rheology there, which were likely caused by lithospheric detachment[52,53]. Small-volume Cretaceous–Paleogene basalts scattered in the Tianshan[54] and contemporaneous high-heat flux (~80–85 mW/m²) inferred from xenoliths[55] suggest that the uppermost mantle in the Tianshan should have been hot and weak before the Cenozoic shortening. Although our Vs model has a limited resolution in the mantle depth, it does image upper mantle low-$V_S$ anomalies overlain by a thin mantle lid extending from the Moho to 80–90-km depth beneath the STAC and the NA (Fig. 2d). A similar mantle lid structure is also indicated in a previous S-wave receiver function study[56]. Such a pre-weakened lithosphere of the Tianshan weaker than its surroundings is plausible to accommodate the vast contractional stresses[2,10,57]. The thin mantle lid also might be beneficial for the Moho thrust deformation, as indicated by the Moho doublet developed in the northern edge beneath the NL (Fig. 2c, d).

Therefore, with the direct contact between the cratonic Indian slab and the Tarim Craton, the India-Asia collisional stress could effectively reach the pre-weakened Tianshan and contribute to its rejuvenation (Fig. 3). Our refined crustal model well demonstrates that inherited properties in the crust controlled shallow brittle deformation as indicated by the pure-shear shortening of the accretionary wedge (STAC) and the thick-skinned deformation of the Paleozoic volcanic arcs (NA and IKA). The plastic deformation accommodated N–S shortening in the deep and thrust-imbricated the Moho into several segments across the central Tianshan. The balanced cross-section of Moho discontinuities supports the idea that intracontinental deformation in the Tianshan intensified synchronously at the time of the direct contact between the cratonic Indian slab and the Tarim Craton in the Late Miocene (~10 Ma). These findings provide a robust and unified rejuvenation model for the Tianshan Orogen, and lead to promising implications for depth-dependent deformation in intracontinental orogens worldwide.

## Methods
**Quality control and harmonic analysis of receiver functions**. We collected data from the MANAS network (40 stations)[25] and three other stations close to the MANAS network to construct a ~475-km long cross-section with a ~10-km station space across the central Tianshan (Fig. 1a, b). We selected events with Mw ≥ 5.0 at an epicentral distance of 30–90° (Fig. 1c) to calculate P-wave receiver functions (RFs) with Gaussian coefficients of 1.0, 2.0, and 3.5 by using the time-domain iterative deconvolution method[58]. After normalizing both time and amplitude of the RFs to a reference slowness of 6.4 s/° (moveout correction[59]), all corrected RFs for each station were stacked in bins of 30° back-azimuth within a delay time window of −5–15 s that include Ps phases converted at the Moho (Pms) but exclude the subsequent multiples (PpPms and PsPms + PpSms). We selected reliable RFs according to the quality control procedure proposed by Shen et al.[60] and

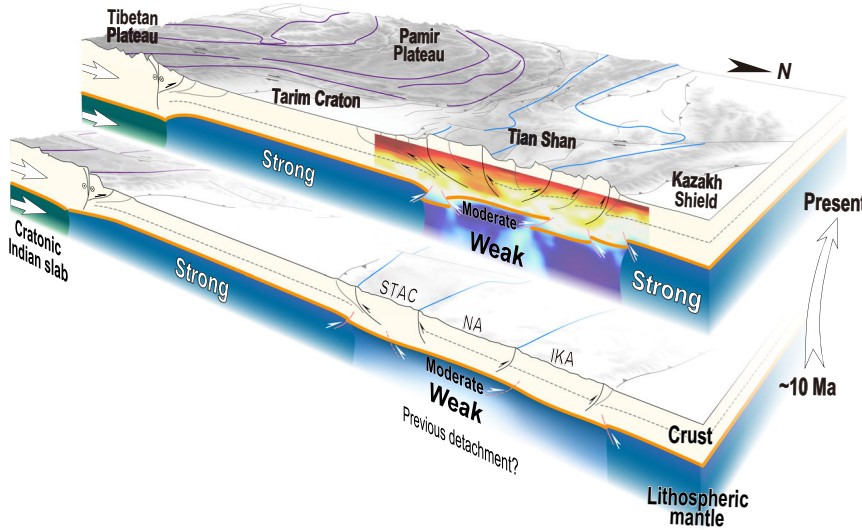

**Fig. 3 Evolutionary profiles illustrating the rejuvenation of the Tianshan Orogen since ~10 Ma.** An earlier detachment of the lithospheric root led to a weaker lithosphere in the Tianshan than in the surroundings. When the cratonic Indian slab directly impinged on the Tarim Craton, the contractional stress from the India-Asia convergence was effectively delivered to the Tianshan. Consequently, the lithosphere of the Tianshan underwent intense deformation, which was accommodated by inherited, structurally controlled brittle deformation in the shallow crust and by plastic deformation close to the Moho. IKA Issyk Kul Arc, NA Naryn Arc, STAC South Tianshan Accretionary Complex.

discarded the following traces: (1) with unrealistic amplitudes (out of 0 ~ 1) at zero time; (2) whose products of convolution with the vertical component have a poor fit with the radial component (<80% recovery of the observed radial seismogram); (3) that were significantly different from the corresponding bin-averaged RF (cross-correlation coefficient <0.5). After the quality control process, a reliable, successive RF dataset was established (Supplementary Fig. 1a–c).

We applied the harmonic analysis method[60] to extract the azimuthally independent RFs with minimal potential influences of tilting interface and azimuthal anisotropy (Supplementary Fig. 1d–h). The piercing points of the *Ps* phases at a depth of 60 km sample a circular area with a diameter of 0.5° centered on each station (Fig. 1b), similar to the horizontal grid scale of the Rayleigh wave dispersion (DPs) (0.5° × 0.5° grids) used in this study[61]. However, the harmonic analysis method may be less effective when the azimuthal coverage of RFs is poor, due to the limits of the distribution of events and data quality. To minimize the bias in estimating the average RF for stations with uneven azimuthal coverage, we calculated bin-averaged RFs within a bin of 30° back-azimuth, and then stacked them to estimate the average RFs and corresponding uncertainties. Subsequently, the average RFs were used in the joint inversion with the DPs. Supplementary Fig. 2b–d present the average RFs with Gaussian coefficients of 1.0, 2.0, and 3.5 for each station, and the *Pms* phases are traceable continuously along the cross-section.

**Joint inversion of receiver function and surface wave dispersion.** Based on the results of the ambient noise tomography (0.5° × 0.5° grids) in the Tianshan and the Pamir Plateau[61], we interpolated group (6–40 s) and phase velocity (6–42 s) DPs at each station (Supplementary Fig. 2e, f). For the central Tianshan, checkerboard tests indicate that group and phase velocity DPs can well resolve anomalies with a size of 0.5° × 0.5° in the shorter periods and 1.0° × 1.0° in the longer periods (Supplementary Fig. 3a), and posterior errors are <2% for all periods (Supplementary Fig. 3b). In this study, we performed a joint inversion of the interpolated DPs and average RFs to obtain a one-dimensional (1-D) $V_S$ model at each station using the Computer Programs in Seismology (CPS, version 3.30) software package[62] that is based on an iteratively damped least-squares algorithm[35]. The $V_S$ was set to a constant value of 4.5 km/s in the initial model. We used the smoothed crustal $V_P/V_S$ ratio for each station by averaging the recent H-κ-c stacking results[27] within a distance of 50 km from each station, and set the $V_P/V_S$ ratio in the upper mantle corresponding to the AK135 model[63]. The density ($\rho$) was calculated as a simple approximate from the Nafe-Drake empirical relationship:[64]
$\rho = 1.73 + 0.2V_P$. The recovery tests using a similar dataset suggest that our datasets and inversion schemes can obtain a robust result down to 120-km depth[61].

As shown in Supplementary Fig. 4, the final 1-D $V_S$ models are well consistent with the interfaces, especially the Moho, as indicated along raypaths in the individual RFs plots and by the single station CCP stacking. The synthetic RFs and DPs calculated from the final 1-D $V_S$ models of the joint inversion fit well with the observed ones. To evaluate the errors of the final model, we used resampled DPs and RFs as input to the same inverse procedure described above for 200 times. The resampled DPs and RFs were generated by adding Gaussian random noise with a standard deviation corresponding to the posterior error (Supplementary Fig. 3b) and the difference between the raw RFs and harmonic RFs (Supplementary

Fig. 1d), respectively. The average values of the 200 $V_S$ models are similar to the final 1-D $V_S$ model with small uncertainties (standard deviation <1%) at depths shallower than 120 km (Supplementary Fig. 4).

The 1-D $V_S$ models at all stations inverted by the above-mentioned scheme (Supplementary Fig. 2g) were assembled to construct the final two-dimensional (2-D) $V_S$ model along the cross-section. A most prominent feature resolved by our $V_S$ model is the contrasting crustal structure across the Tianshan orogen, i.e., an extensive low-$V_S$ (~3.4 km/s) middle-lower crust beneath the STAC, whereas low-$V_S$ anomalies (~3.5 km/s) in the middle-lower crust confined within several bands beneath the NA and the IKA (Fig. 2d). Three elongate low-$V_S$ anomalies, which can be easily recognized in the 2-D $V_S$ model beneath stations KARD, BESM, and SOUR (Fig. 2d), are located at depths of ~20–50 km as seen in the 1-D $V_S$ models, respectively (Supplementary Figs. 4b–d). To test the robustness of the low-$V_S$ anomalies, we modified the 1-D $V_S$ models of stations KARD, BESM, and SOUR, by excluding the crustal low-$V_S$ anomalies, and calculated synthetic RFs and DPs. We find significant differences between the synthetic and observed DPs, and the synthetic RFs from the modified models fail to fit the negative *Ps* phases at ~3–4 s (Supplementary Fig. 5a–c). These large mismatches between the observed and calculated RFs and DPs indicate the reliability of the low-$V_S$ anomalies in our $V_S$ model. We also performed robustness tests of the mantle lid structure imaged beneath the STAC and the NA (Fig. 2d). Both synthetic DPs from the final and the modified 1-D $V_S$ models are similar to the observed ones, but the synthetic RFs from the modified models, by excluding the high-$V_S$ mantle lid at depths of ~50–80 km, fail to fit the *Ps* phases at ~6–8 s (Supplementary Fig. 5d, e). These features indicate that the mantle lid structure is also reliable although DPs used in this study have a limited resolution of the mantle structure.

**Common conversion points stacking of receiver functions.** Referring to the final 1-D $V_S$ model at each station, we projected the reliable RFs into the depth domain, and use the CCP stacking technique[36] to image the geometry of the Moho and intracrustal interfaces. Due to the $V_P/V_S$ ratio being fixed during the joint inversion, the $V_P/V_S$ ratios at each station used in the CCP stacking are the same as we set in the initial model of the joint inversion, which are smoothed from previous H-κ-c stacking results in the crust[27] and correspond to the AK135 model in the upper mantle[63]. We constructed CCP stacked RF images with different Gaussian parameters (i.e., 1.0, 2.0, and 3.5) to distinguish a true interface, which should be consistent in different frequency bands. The Moho is visible as a strong positive *Ps* phase in all three CCP images with depths between ~45 km and ~70 km (Supplementary Fig. 6). Referring to the same velocity model at each station, we also constructed CCP images using multiples (i.e., *PpPs* phases) that reverberated between the free surface and interfaces in the crust and upper mantle. Given the fact that the *PpPs* phases exhibit weaker sensitivities to the $V_P/V_S$ ratio than the *Ps* phases, the consistency in Moho depths of the CCP images constructed with the *PpPs* phases and the *Ps* phases confirms the reliability of the CCP images and indicates the suitability of the $V_P/V_S$ ratio used in the inversion of the $V_S$ model (Supplementary Fig. 7).

Besides the Moho, our CCP images constructed with *Ps* phases also reveal clear but intermittent intracrustal signals (Supplementary Fig. 6). The marked intracrustal

signals are traceable in the image of individual RFs traces and in the CCP images with different stacked-bin parameters (2.5, 5.0, and 7.5 km), which largely excludes the possibility that the intracrustal signals are derived from noise or scattering (Supplementary Fig. 8). Moreover, the CCP images produced by a fraction (20% and 60%) of reliable RFs are consistent with those made by the entire dataset, implying that the effect caused by an uneven data distribution or anisotropy structure is negligible (Supplementary Fig. 9). We undertook some synthetic tests to check the reliability of the intracrustal Ps phases. We constructed the synthetic model at each station by overlaying the final 1-D model of $V_S$ and $V_P/V_S$ ratio at depths shallower than 10 and 20 km, respectively, on a homogeneous half-space model (Supplementary Fig. 10b, d). Then, we synthesized RFs with the same data coverage as the real dataset, and constructed synthetic CCP images by using the same processing parameters as used for the real dataset (Supplementary Fig. 10c, e). The significant negative Ps phases at depths of 10–20 km beneath the Tarim Basin and Kazakh Shield are clearly generated by the effect of the sedimentary layer, but not by real velocity discontinuities. Except for these artifact signals in the northern and southern ends of the cross-section, synthetic CCP images keep their purities across the Tianshan. These tests strongly suggest that the marked intracrustal Ps phases (Fig. 2c, d; Supplementary Figs. 6–9) in the CCP images are robust. Thus, the discussion and interpretation in this study focused on these reliable features mostly in the Tianshan, but not at the ends of the cross-section.

**Balanced cross-section estimates of N–S shortening across the Tianshan.** The $V_S$ model and CCP images collectively reveal a thrusting Moho geometry along the cross-section (Fig. 2c, d). Because the peneplanation in the Tianshan is generally inferred to have developed during the Mesozoic tectonic quiescence[44], the Moho segments and thrusts imaged in this study can be used as a marker for estimating the amount of N–S shortening across the Tianshan in response to the India-Asia collision in the Cenozoic. Assuming the Moho beneath Tianshan was flat before the Cenozoic intracontinental deformation, we estimate that the N–S width before the deformation of the imaged region was ~560 km by summing the lengths of each Moho segment, which indicates ~85-km N–S shortening when considering the ~475-km width at present (Fig. 2e). It should be noted that a balanced cross-section of the Moho length should represent the minimum estimate because of the limited resolution of seismic images on some small-scale thrusts or doublets. Furthermore, we assume that the crustal volume remained constant during the Cenozoic deformation, and that the Tianshan had a ~40-km thick crust before the Cenozoic, as indicated by the P-T equilibration conditions of xenoliths[49] and by the present-day Moho depth beneath the Tarim Craton and the Kazakh Shield (Fig. 2c, d). The present-day crustal area of $~2.42 \times 10^4$ km² imaged along the 2-D cross-section implies a crustal dimension of ~605-km N–S width and ~40-km thickness before the Cenozoic deformation, and thus indicates ~130-km N–S shortening across the Tianshan in the Cenozoic. Given the variation of previous crustal thicknesses and the erosion of the uplifted area, the N–S shortening estimated by balancing the crustal thickening also has some uncertainties. We consider these balanced cross-section estimates as a first-order indication of the N–S shortening of a hundred kilometers across the Tianshan. Consequently, considering the present-day N–S shortening rates of ~12–15 mm/yr suggested by Late Quaternary fault slip rates and GPS observations across the estimated section (Fig. 2a, b), the N–S shortening of ~85–130 km implies a start time of ~11–6 Ma for the significant N–S shortening across the Tianshan.

## Data availability

The data of the seismic stations used in this study are available through the Data Management Center of the Incorporated Research Institutions for Seismology (IRIS DMC), stations from network MANAS (XP) at https://doi.org/10.7914/SN/XP_2005; station ARLS from network KR at https://doi.org/10.7914/SN/KR; and stations TERE and TGMT from network XW at https://doi.org/10.7914/SN/XW_1997.

## Code availability

The Computer Programs in Seismology (CPS, version 3.30) software package is available from the Saint Louis University Earthquake Center (http://www.eas.slu.edu/eqc/eqccps.html). The common conversion points (CCP) stacking code is available from the author, Prof. Lupei Zhu, at his homepage http://www.eas.slu.edu/People/LZhu/home.html.

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

## Acknowledgements

This research was supported by the Strategic Priority Research Program (B) of the Chinese Academy of Sciences (grant XDB18000000), the National Natural Science Foundation of China (grants 41888101 and 41804056), and the National Key R&D Program of China (grants 2016YFC0600402 and 2017YFC0601206). W.L. was also supported by the China Postdoctoral Science Foundation (grants 2019M650034 and 2020T130650) and the Sino-German (CSC-DAAD) Postdoc Scholarship. This is a contribution to IGCP 662 and 669.

## Author contributions

W.L., in interaction with Y.C. and X.Y., collected and processed the seismic data. W.X. and B.F.W. shaped the tectonic implications of the study. W.X., Y.C., and W.L. developed the initial interpretation. All authors contributed to the writing of the paper.

## Competing interests

The authors declare no competing interests.
