## [Peer Review File · Nature Communications]

Intracontinental deformation of the Tianshan Orogen in response to India-Asia collisionREVIEWER COMMENTS

Reviewer #1 (Remarks to the Author):

See the attached PDF file.

This is my review of the manuscript entitled “**Intracontinental deformation of the Tianshan Orogen in response to India-Asia collision**” by Li et al. The manuscript shows very high-resolution and interesting crustal images that provide new understanding of how the Tianshan crust and mantle lithosphere has responded to the underthrust/subduction of the Indian plate at the far plate boundary. Although there are remaining questions (as pointed out by the authors), the presented work has very important implication about our understanding of intracratonic lithosphere deformation, particularly in response to far-field plate boundary forces/processes. The manuscript is well organized and well-written. My comments are mostly on the technical and methodological sides. The manuscript could be improved by addressing/clarifying these points.

Major comments/questions:

1. Regarding the Moho doublet beneath the NL, if there is an overthrust northward at the Moho, I would expect to see a negative CCP amplitude on top of the lower Moho segment (the footwall segment). Here the image only shows two positive conversions corresponding to the broken Moho. Is this a resolution issue or there are other reasons?
2. The dipping (particularly south-dipping) low-velocity strips in the mid-lower crust are very interesting observations. Disregarding the overlain interpretation (fault planes/arrows), they are relatively weak velocity anomalies and in relatively small (~10 km horizontal scales) scales. A common question for seismic imaging like this would be whether they can be resolved with the data and the method? The S-wave velocity model was constructed by joint inversion of receiver function and Rayleigh wave dispersions. When comparing the CCP image and the S-wave model, since they are not totally independent results, I am wondering whether some of these dipping features were introduced by receiver function artifacts, e.g., free surface and shallow crustal multiples.
3. Related to the above point, there is no low-velocity strip/dipping zone in the northern end of the image corresponding to the strong negative dipping interface below the Kazakh Shield and part of the IKA. At least, I would expect a strong low-velocity lower-crustal layer below this region. Supplementary Figure 5 shows that this negative feature persists at all frequency bands. Do the authors have any explanations for this? In addition, the Vs model beneath the STAC is rather homogeneous, though the CCP image shows a clear negative anomaly/conversion. This might be related to the resolution of the Vs model as pointed out in the item below. Any thoughts?
4. Maybe it is due to the color scale, I don't see any notable variation of velocity at the top 20-30 km in the S-wave velocity model. The CCP image however shows clear segmentation full of thrusting structures. Is this a resolution issue for the S-wave model?
5. The authors used Rayleigh waves down to about 40 seconds but the velocity model shows images down to 120 km. A 40-second Rayleigh wave is most sensitive to the depth of 30-60 km (<https://academic.oup.com/gji/article/188/1/131/632499>). Does this limit the resolution (and therefore interpretation) of the mantle lithosphere? This is important considering the observation of the thin mantle lid below the STAC and NA regions and the low-velocity weak mantle lithosphere below the lid.

Xiaotao Yang

Reviewer #2 (Remarks to the Author):

Comments on the manuscript entitled “Intracontinental deformation of the Tianshan Orogen in response to India-Asia collision” by Wei Li et al.

The reactivation of the Tian Shan in Cenozoic is generally regarded as a result of the long-lived and still-ongoing India–Asia collision. Many issues are still hotly debated; for example, how the plate-convergent stress is transferred from the plate boundary to the far-field Asian interior and how the east-west-elongated Tian Shan accommodates the convergence.

In this work, the authors present new seismic images of a lithospheric cross-section of the Tian Shan, based on receiver functions and Rayleigh wave dispersions along a N–S-trending linear seismic array. An extensively deformed lithosphere, especially crust, is observed, with illustrating the brittle structures in the shallow crust and plastic deformation near the Moho. On the other hand, the presented seismic images record only limited underthrusting of the Tarim and Kazakh blocks beneath the Tian Shan. Together with the balanced cross-section restoration of the Moho length and crustal volume, the authors propose that the pre-weakened lithosphere and effective stress transfer are required for the Cenozoic rejuvenation of the Tian Shan. The seismic images to some extent reflect the weakness of the Tian Shan, and illustrate how the crust deformed in response to the plate convergence. The manuscript is clearly and well written.

My specific comments and suggestions are shown below:

1. The current seismic profile covers only the Tian Shan orogen, whereas a large ‘picture’ is illustrated, for example, with the relationship to the far-field India-Asia collision and stress transfer. I would thus suggest elaborating the discussion on the linkage between the detailed Tian Shan deformation styles and the India–Asia collision, which may further increase the significance of this study.

2. Following the first suggestion, this seismic study tends to support the inference that the India–Tarim collision/contact beneath West Kunlun dominates the Cenozoic reactivation of the Tian Shan, as proposed by a recent numerical modeling study (Huangfu et al., 2021). Besides that, there are two other models that were commonly invoked to account for the extensive deformation in the Asian interior. One attributes the uplift of the Tian Shan to a progressive building of the Tibetan plateau due to the India–Asia collision. Eventually, its accumulated high potential energy allows transmission of the stress across the Tarim block to reactive the Paleozoic Tian Shan orogenic belt (see papers by Peter Molnar and/or Paul Tapponnier). Alternatively, another model assumes that the initiation of the Cenozoic deformation of the Tian Shan was a consequence of the hard collision between Arabia and Eurasia, which generated a series of strike-slip faults from the Zagros to Lake Baikal. The termination of these right-slip faults

produced the Tian Shan (see Yin An, 2010). I understand that the scale of the present seismic observations may be not enough for ruling out these additional models, but relevant discussion should be necessary. In particular, what kind of constraints, if there are any, can be provided by this study on clarifying these contrasting models as well as better understanding the intra-continental orogeny.

3. An interesting point of this seismic observation is the distinct crustal deformation patterns of the southern part of Tian Shan (STAC), comparing to the middle and northern part (NA–IKA). The former one is more like pure-shear thickening with diffusive deformation and low strain rate, whereas the latter one is characterized by the thick-skinned thrusting with imaging several thrust faulting zones (Figure 2). The authors attribute this phenomenon to the contrasting rheological strength of the different terranes: “Such an accretionary wedge (STAC) was too weak to deliver the stresses and has continuously shortened in response to the India-Asia collision, whereas the crust of the Paleozoic volcanic arcs (NA and IKA) to the north was so strong that active faults were able to penetrate into the deep crust to accommodate the contractional stress (Figs. 2b–d).” I generally agree with this inference; but, it is better to add some previous observations, for example, the electrical structures, which may verify this inference in a certain degree.

4. With the observed distributed deformation across the Tian Shan, the authors argue against the underthrust of Tarim and Kazakh blocks beneath the Tian Shan. It is a bit puzzling about this point. From the seismic images, the Moho in the northern (IKA) and southern (STAC) parts of Tian Shan is deeper than that in the middle (NA); will this phenomenon give some implications for the underthrusting? Please discuss and clarify it.

5. The Tian Shan has a complex deep structure along strike. A wealth of previous geophysical studies revealed contrasting along-strike crustal–lithospheric structure of the Tian Shan, in terms of the crustal thickness, Moho and LAB morphology, and lithospheric interplay with the adjacent blocks. In particular, distinct deformation styles were suggested to regulate different section of the present-day Tian Shan, e.g., compression in central Tian Shan but trans-tension in eastern Tian Shan. Although this study is focusing the western Tian Shan, I am wondering whether the current data and model can be compared with the central and eastern Tian Shan, in order to give a comprehensive impression about the Cenozoic building of this great intra-continental orogen, under the far-field effects of India-Asia collision, as well as the possible rotation of the strong Tarim Block (Zhao et al., 2019; <https://doi.org/10.21285/2686-9993-2019-42-4-425-436>). I understand this may be challenging, thus, just provide as a (future) suggestion for the authors.

Zhong-Hai Li

Reviewer #3 (Remarks to the Author):

In this study, the authors present a 2-D Vs model and a CCP stacked receiver function image of a N-S section across the Tianshan Orogen. The seismic data were recorded by a dense linear array deployed across the orogen. While I like the careful work that the authors have done in producing the receiver function image and in obtaining the 2D S-wave model, I am not so sure whether the nice seismic results can be used here to address the scientific question on the timing of intracontinental deformation of the Tianshan Orogen, as well as whether the deformation is the response to the India-Asia collision. It is also true that you can speculate that the Tianshan orogen has a pre-weakened lithosphere before its rejuvenation in Cenozoic, but it is not a direct conclusion of your CCP image and Vs model. In conclusion, the scientific question of this study is not well phrased at the current draft, therefore I would recommend a re-submission so the authors can revise the manuscript by clearly identifying a scientific question that can be directly addressed by the seismic data here.

My detailed comments and questions for the authors are included below.

Major points

1. You stated that “How the continental lithosphere deforms far away from plate boundary has been long debated” without saying anything about the debate and current hypotheses. It is also unclear what are the exact scientific questions on the intracontinental deformation of the Tianshan orogen that the authors want to address: the cause and timing of the Cenozoic rejuvenation, deformation style (continuous or episodic) and deformation mechanism (due to underthrust of the Tarim and Kazakh lithosphere or whole crustal/lithosphere shortening), etc. You need to provide the necessary background information on the current understanding of these questions and present a clear picture that these questions/hypotheses can be directly answered by seismic data. Same is also true for the main conclusions. Is the 10-Myr estimate of deformation history one of the main conclusions here? Has it been suggested by previous geological and/or geochemical studies? Also are there any evidence suggesting that the deformation rate is roughly constant over this 10-Myr period? Also, a pre-weakened lithosphere underneath the orogen is a speculation, not a direct result of the seismic data.
2. The authors show a schematic figure on balance section calculation (Figure 2f) without giving any details in the method. I think this observation is a key component of this study, it is better for the authors to provide more details in the method section on how to compute the N-S shortening using the identify Moho segments and crustal thickness. It seems that there are 6 Moho segments (dashed lines) shown in Figures 2c and 2d. Can you list the length of each segment, so the readers can understand where the number of ~85 km shortening comes from? Same is for the crustal volume. The two estimates are quite different (~85 km versus ~130 km), what are the likely causes of this difference?
3. I think that the authors need to provide more details on the data and parameterization either in the method section or in the supplementary document. What is the station spacing of the combined linear

array? What is the grid spacing here ($475 \text{ km}/200 \text{ grid} = 23.75 \text{ km}$)? It seems that the phase/group velocity data are from a previous tomography study. What is the spatial resolution of the phase/group velocity maps? What are the uncertainties in the phase/group velocity data?

4. Lines 418-420: It is unclear to me why the authors can claim that PpPs phases have weaker sensitivities to the velocity structure than the Ps phases. Also, what is the V_p/V_s ratio of each 1-D V_s profile used in the computing Ps and PpPs offsets? Do you use the same V_p/V_s ratio for all the profiles?

5. The cartoon in Figure 3 shows the Indian slab underthrusts almost horizontally to the north and collides with the Tarim cratonic mantle lithosphere. I am not sure whether this is the case at the location of the N-S profile, as the authors mention that the India slab is imaged beneath the Pamir Plateau in the west (Figure 1a). Also, the India slab shows an obvious northward dipping indicated by the intermediate seismicity (Figure 1a), so it is not underthrust horizontally. More importantly, the subducted India slab is likely an oceanic lithosphere, much weaker than a cratonic lithosphere. So, I am not sure whether this cartoon is correct here.

Minor points

6. Line 112: "However, the large amount of shortening ($>500 \text{ km}$)..." It is unclear where this number comes from. The authors need provide more details.

7. Figure 2a: It seems that the strain rate and GPS motion data shown in Figure 2a are not so consistent. The GPS data suggest that deformation is roughly uniformly distributed across the orogen, while the strain rates indicate a northward decreasing with spikes not necessarily collocated with faults and sutures.

8. I find some inconsistency between the CCP image, and the V_s model shown in Figures 2c and 2d, respectively. For example, at -60 km , the velocity jump associated with the deeper Moho is not so clear. Also at $80\text{-}120 \text{ km}$, a strong negative dipping event at $10\text{-}20 \text{ km}$ is shown in the CCP image, but you don't see a large downward velocity drop at the same depths in the V_s model? For the example station BESM shown in the Supplementary figure 4, it is better to divide the individual receiver functions recorded at this station into two groups: one from south and the other from north. As the station is located at the boundary of a Moho step, you would expect to see a significant difference in Pms between the two groups.

9. Figure 2e, when you compute the crustal shortening using crustal volume, I assume you have considered the elevation. But how do you take erosion into account?

10. Line 383: V_p/V_s ratio of the upper mantle is 1.75. Is it too low? IASP91/AK135 has a V_p/V_s ratio of ~ 1.80 . Will this assumption affect your inverted V_s ?

Response to Reviewer #1:

This is my review of the manuscript entitled “Intracontinental deformation of the Tianshan Orogen in response to India-Asia collision” by Li et al. The manuscript shows very high-resolution and interesting crustal images that provide new understanding of how the Tianshan crust and mantle lithosphere has responded to the underthrust/subduction of the Indian plate at the far plate boundary. Although there are remaining questions (as pointed out by the authors), the presented work has very important implication about our understanding of intracratonic lithosphere deformation, particularly in response to far-field plate boundary forces/processes. The manuscript is well organized and well-written. My comments are mostly on the technical and methodological sides. The manuscript could be improved by addressing/clarifying these points.

R: Thank you for all the constructive comments. By carefully considering these comments, we revised the manuscript point-by-point. We provided more details about the data and method, and carried out some new synthetic tests to assess the resolution ability of this study. Please see the following for more details. Note please, the line numbers listed in the responses are referred to the manuscript with tracking changes (*the attached file Manuscript.NC.R1_TrackingChanges.pdf*).

Major comments and questions:

1. *Regarding the Moho doublet beneath the NL, if there is an overthrust northward at the Moho, I would expect to see a negative CCP amplitude on top of the lower Moho segment (the footwall segment). Here the image only shows two positive conversions corresponding to the broken Moho. Is this a resolution issue or there are other reasons?*

R: This is mainly a resolution issue. We calculated synthetic RFs with different Gaussian parameters for different 1-D Moho doublet models to check the resolving ability of the negative P_s phase corresponding to the velocity decreasing below the upper Moho discontinuity. In these models, the structure between the two overlapping Moho interfaces is represented by a range of velocity drops from a sharp contrast to a gradient variation. As shown in the figure below, the amplitude of the negative P_s phase is related to the decreasing gradient of velocity. A sharp velocity contrast may be seen by high-frequency RFs, but a gradient velocity variation will not be well resolved in all frequency bands. For a plastic deformation close to the Moho, the fault plane might be not so sharp to be recognized as a negative P_s phase in the receiver function. Possible dipping fault planes or inhomogeneities also make it is difficult to resolve a clear negative P_s phase between two positive P_s phases in only a 20-km-thick layer. However, for the case in this study, we can trace the upper and lower Moho as strong positive P_s phases separately to the south and to the north (Fig. 2c), which indicates the thrust discontinuity is a reasonable interpretation.

2. The dipping (particularly south-dipping) low-velocity strips in the mid-lower crust are very interesting observations. Disregarding the overlain interpretation (fault planes/arrows), they are relatively weak velocity anomalies and in relatively small (~10 km horizontal scales) scales. A common question for seismic imaging like this would be whether they can be resolved with the data and the method? The S-wave velocity model was constructed by joint inversion of receiver function and Rayleigh wave dispersions. When comparing the CCP image and the S-wave model, since they are not totally independent results, I am wondering whether some of these dipping features were introduced by receiver function artifacts, e.g., free surface and shallow crustal multiples.

R: Thank you for the constructive comment. We carried out some synthetic tests to illuminate the reliability of these crustal LVZ observed in the CCP image and V_S model. At first, we modified the 1-D V_S model beneath stations KARD and BESM by excluding the crustal low- V_S anomalies and calculated synthetic RFs and DPs (Lines 335–341, Supplementary Fig. 5). The synthetic DPs are significantly different from the observed ones, and the synthetic RFs failed to fit the negative P_s phases at 3–4 s. These large mismatches indicate that the low- V_S anomalies are necessary for the V_S models. Secondly, we calculated synthetic RFs for the shallow parts of the final V_S model, shallower than 10- and 20-km depth, respectively, and constructed synthetic CCP images using the same processing parameters as used for the real dataset (Lines 366–375, Supplementary Fig. 10). Except for the artifact signals in the north and south ends of the profile, synthetic CCP images keep their purities across the Tianshan. These tests strongly suggest that the marked intracrustal P_s phases in the CCP images are robust features, and thus the discussion is mainly focused on the structures in the Tianshan but not the ends of the cross-section (Lines 375–378).

3. Related to the above point, there is no low-velocity strip/dipping zone in the northern end of the image corresponding to the strong negative dipping interface below the Kazakh Shield and part of the IKA. At

least, I would expect a strong low-velocity lower-crustal layer below this region. Supplementary Figure 5 shows that this negative feature persists at all frequency bands. Do the authors have any explanations for this? In addition, the V_s model beneath the STAC is rather homogeneous, though the CCP image shows a clear negative anomaly/conversion. This might be related to the resolution of the V_s model as pointed out in the item below. Any thoughts?

R: As mentioned in the responses above, we added new synthetic tests to check the effect of shallow structures in the CCP images. In the synthetic CCP images (Supplementary Fig. 10), we can see significant negative P_s phases at depths of 10–20 km beneath the Tarim Basin and Kazakh Shield which are artifacts and definitely generated by the effect of the sedimentary layer. Our V_s model also shows normal velocities at the depths below artificial these negative P_s phases.

Below the negative P_s phase at the depth of ~30 km beneath the STAC, we indeed find a slightly low-velocity zone and interpreted as the northern termination of the shallow detachment faults in the Tarim Basin (Fig. 2d). Because the station distribution beneath the STAC is not as uniform as other parts along the profile, the extended range of negative phases in the shallow depths may be caused by smearing in the CCP stacking (Supplementary Fig. 8b).

Moreover, DP and RF have different resolution ability, the former is more sensitive to absolute V_s and the latter provides more constraints on the velocity discontinuities. Although the harmonic analysis or bin-average stacking can minimize the effects of tilting interface and azimuthal anisotropy to obtain azimuthally independent RFs coordinated with the DPs at each station (Lines 285–294, Supplementary Fig. 1), it's difficult to completely resolve the inconsistency between these different datasets. Thus, we interpreted the CCP images and V_s model obtained in this study together with previous geological and geophysical results to avoid misinterpretations that could easily be caused by using single information.

4. *Maybe it is due to the color scale, I don't see any notable variation of velocity at the top 20-30 km in the S-wave velocity model. The CCP image however shows clear segmentation full of thrusting structures. Is this a resolution issue for the S-wave model?*

R: We observed some variations of the shallow structures (0–20-km depth) both in the V_s model and CCP images. As shown in Fig. 2c and 2d, we marked some discontinuities at depths of ~20 km beneath the NA and IKA, which are corresponding to positive P_s phases and the V_s increases from ~3.5 km/s to ~3.7 km/s there and likely represent the Conrad discontinuity between the upper and lower continental crust (Lines 39–45 in Supplementary Information). These discontinuities are clearly tilting and separated by the interpreted faults extending from deep crustal low- V_s bands to the shallow. As shown in Supplementary Fig. 10, there are also some notable variations in the V_s model shallower than 10-km depth, which mainly reflect the characters of the sedimentary layer.

5. The authors used Rayleigh waves down to about 40 seconds but the velocity model shows images down to 120 km. A 40-second Rayleigh wave is most sensitive to the depth of 30-60 km (<https://academic.oup.com/gji/article/188/1/131/632499>). Does this limit the resolution (and therefore interpretation) of the mantle lithosphere? This is important considering the observation of the thin mantle lid below the STAC and NA regions and the low-velocity weak mantle lithosphere below the lid.

R: As the reviewer said, the group (6–40 s) and phase velocity (6–42 s) DPs used in the inversion are most sensitive to the crustal structures and have limited resolution in the upper mantle. In this study, we performed joint inversion of these DPs and the RFs within a delay time window of –5 to 15 s, thus, these data can provide some more constraints in the upper mantle. Please see the figure below for a simple recovery test to assess the resolving ability of anomalies in the crust and upper mantle, Li et al.⁶⁷ (2020) constructed this test with similar datasets and inversion schemes as in this study except for the inclusion of the RF with a Gaussian parameter of 3.5. We can see that accurate V_S and depth ranges of the anomalies in the crust and upper mantle can be well recovered by the joint inversion of DPs and RFs. To evaluate the errors of the final V_S model, we resampled the DPs and RFs by adding Gaussian random noise corresponding to the observation errors (Supplementary Figs. 1d and 3b) and inverted them as input to the same inverse procedure. The average values of the V_S models obtained in the resampling tests are similar to the final 1-D V_S model with small uncertainties (standard deviation less than 1%) at depths shallower than 120 km (Supplementary Fig. 4). These tests suggest that the joint inversion scheme driven by our datasets can work robustly for deep structures to 120-km depth (Lines 311–314, 318–326).

Response to Reviewer #2:

Comments on the manuscript entitled “Intracontinental deformation of the Tianshan Orogen in response to India-Asia collision” by Wei Li et al.

The reactivation of the Tian Shan in Cenozoic is generally regarded as a result of the long-lived and still-ongoing India–Asia collision. Many issues are still hotly debated; for example, how the plate-convergent stress is transferred from the plate boundary to the far-field Asian interior and how the east-west-elongated Tian Shan accommodates the convergence.

In this work, the authors present new seismic images of a lithospheric cross-section of the Tian Shan, based on receiver functions and Rayleigh wave dispersions along a N–S-trending linear seismic array. An extensively deformed lithosphere, especially crust, is observed, with illustrating the brittle structures in the shallow crust and plastic deformation near the Moho. On the other hand, the presented seismic images record only limited underthrusting of the Tarim and Kazakh blocks beneath the Tian Shan. Together with the balanced cross-section restoration of the Moho length and crustal volume, the authors propose that the pre-weakened lithosphere and effective stress transfer are required for the Cenozoic rejuvenation of the Tian Shan. The seismic images to some extent reflect the weakness of the Tian Shan, and illustrate how the crust deformed in response to the plate convergence. The manuscript is clearly and well written.

R: We really appreciate the reviewer’s overview of the study and constructive comments. In the revised manuscript, we gave a more comprehensive introduction of the background and regional tectonics which will help to illuminate the scientific issues of this study. We also compared our results with previous studies in more details to confirm the reasonability of interpretation. Please see the detailed replies to comments and suggestions below. The line numbers listed in the responses are referred to the manuscript with tracking changes.

Specific comments and suggestions:

1. The current seismic profile covers only the Tian Shan orogen, whereas a large ‘picture’ is illustrated, for example, with the relationship to the far-field India-Asia collision and stress transfer. I would thus suggest elaborating the discussion on the linkage between the detailed Tian Shan deformation styles and the India–Asia collision, which may further increase the significance of this study.

R: Good suggestion. We rewrote the introduction section to include the debated mechanisms of intracontinental deformation in the Tianshan and point out incoherence interpretations between the shallow deformation and deep structures in previous studies (Lines 39–65). A key to understanding these incoherence interpretations is to establish a detailed image of the spatial relationship between the surface and deep structures. Combining with regional tectonic evolution history, the comprehensive interpretation will have significant implications for how and where the continental lithosphere responds to far-field plate collisions. We hope that the newly added statements will help clarify the significance of this study and make the following discussion more readable.

2. *Following the first suggestion, this seismic study tends to support the inference that the India–Tarim collision/contact beneath West Kunlun dominates the Cenozoic reactivation of the Tian Shan, as proposed by a recent numerical modeling study (Huangfu et al., 2021). Besides that, there are two other models that were commonly invoked to account for the extensive deformation in the Asian interior. One attributes the uplift of the Tian Shan to a progressive building of the Tibetan plateau due to the India–Asia collision. Eventually, its accumulated high potential energy allows transmission of the stress across the Tarim block to reactive the Paleozoic Tian Shan orogenic belt (see papers by Peter Molnar and/or Paul Tapponnier). Alternatively, another model assumes that the initiation of the Cenozoic deformation of the Tian Shan was a consequence of the hard collision between Arabia and Eurasia, which generated a series of strike-slip faults from the Zagros to Lake Baikal. The termination of these right-slip faults produced the Tian Shan (see Yin An, 2010). I understand that the scale of the present seismic observations may be not enough for ruling out these additional models, but relevant discussion should be necessary. In particular, what kind of constraints, if there are any, can be provided by this study on clarifying these contrasting models as well as better understanding the intra-continental orogeny.*

R: Thank you for pointing out the incomplete clarification of the debated mechanisms of the intracontinental deformation in the Tianshan. As mentioned in the responses above, we rewrote the introduction section in more details. Neither the high gravitational potential energy transmission caused by the India-Asia collision nor the transpressional system of the right-slip shears induced by Arabia-Asia oblique convergence can perfectly interpret the uplift of Tianshan. We clarify that these defective hypotheses may show in the inconsistency of the age of deformation or the intensity of continental collision (Lines 39–53). It's important to obtain a detailed configuration of the shallow and deep structures in the Tianshan to understand the debated mechanisms. Recent numerical models¹¹ (Huangfu et al., 2021) propose that the uplift of Tianshan is related to the direct collision between Indian and Tarim lithospheric mantles beneath West Kunlun and can interpret the time lag between the initial India-Asia collision and the Tianshan uplift. In this study, we observe only a hundred kilometers of Cenozoic N–S shortening across the Tianshan, according to the balanced cross-section of Moho discontinuities. These observations not only rule out the large-scale continental subduction of the Tarim Craton or the Kazakh Shield beneath the Tianshan, but also confirm that the intracontinental deformation in the Tianshan intensified synchronously with the direct contact between the Indian slab and the Tarim Craton since the late Miocene (~10 Ma). Together with imaged low- V_S anomalies in the uppermost mantle beneath the Tianshan, this study has an implication for the requisite conditions of the intracontinental deformation, that is pre-weakened lithosphere and effective stress delivery.

3. *An interesting point of this seismic observation is the distinct crustal deformation patterns of the southern part of Tian Shan (STAC), comparing to the middle and northern part (NA–IKA). The former one is more like pure-shear thickening with diffusive deformation and low strain rate, whereas the latter*

one is characterized by the thick-skinned thrusting with imaging several thrust faulting zones (Figure 2). The authors attribute this phenomenon to the contrasting rheological strength of the different terranes: “Such an accretionary wedge (STAC) was too weak to deliver the stresses and has continuously shortened in response to the India-Asia collision, whereas the crust of the Paleozoic volcanic arcs (NA and IKA) to the north was so strong that active faults were able to penetrate into the deep crust to accommodate the contractional stress (Figs. 2b–d).” I generally agree with this inference; but, it is better to add some previous observations, for example, the electrical structures, which may verify this inference in a certain degree.

R: We like the expression of the deformation in the STAC as “pure-shear thickening with diffusive deformation”, and replaced the “continuous shortening” used in the previous manuscript by this phrase (Lines 111, 115, and 123). We cited the results of a magnetotelluric study⁴⁰ (Bielinski et al., 2003) to support our discussion of the contrasting rheological strength of the different terranes in the manuscript (Lines 116–119). The crust of the central Tianshan is composed of several isolated high resistivity bodies separated by more conductive zones in the north, but the southern part adjacent to the Tarim Craton is generally characterized by lower resistivities. These are consistent with our observation and interpretation.

4. With the observed distributed deformation across the Tian Shan, the authors argue against the underthrust of Tarim and Kazakh blocks beneath the Tian Shan. It is a bit puzzling about this point. From the seismic images, the Moho in the northern (IKA) and southern (STAC) parts of Tian Shan is deeper than that in the middle (NA); will this phenomenon give some implications for the underthrusting? Please discuss and clarify it.

R: As stated in the manuscript, our results demonstrate that the high- V_S lower crust of the Tarim Craton does indeed extend northwards with its salient deep Moho, which is consistent with previous observations (Lines 131–136). However, the northward extension of the high- V_S lower crust and the shallow seismic activity suggest that the underthrusting of the Tarim Craton should terminate near the North Tarim Fault (NTF). Given the weak seismic activity in the southern margin of the Kazakh Shield, the southward underthrusting of the Kazakh Shield should be very limited. Please see a detail discussion between our images and previous results in the Supplementary Information Text 1. The shallow brittle deformation generally indicates that the N–S shortening was broadly distributed throughout the Tianshan. This is also inconsistent with the large-scale underthrusting of the Tarim Craton and/or the Kazakh Shield beneath the Tianshan, which would expect the N–S shortenings localized along the southern and northern boundaries of the Tianshan. Thus, we propose the deep deformation of the Tianshan is performed as the Moho segments discontinued by localized thrusts but not the large-scale underthrusting of the Tarim Craton and the Kazakh Shield. These localized thrusts also can thicken the crust in different segments. As shown in the V_S model, the mantle lid beneath the NA is so thin that it can be easily thrust into the adjacent crust without being fragmented, which will

maintain a flat Moho there, although it is a speculation according to the Moho doublet in the northern edge of the NA (Lines 183–187).

5. The Tian Shan has a complex deep structure along strike. A wealth of previous geophysical studies revealed contrasting along-strike crustal–lithospheric structure of the Tian Shan, in terms of the crustal thickness, Moho and LAB morphology, and lithospheric interplay with the adjacent blocks. In particular, distinct deformation styles were suggested to regulate different section of the present-day Tian Shan, e.g., compression in central Tian Shan but trans-tension in eastern Tian Shan. Although this study is focusing the western Tian Shan, I am wondering whether the current data and model can be compared with the central and eastern Tian Shan, in order to give a comprehensive impression about the Cenozoic building of this great intra-continental orogen, under the far-field effects of India-Asia collision, as well as the possible rotation of the strong Tarim Block (Zhao et al., 2019; <https://doi.org/10.21285/2686-9993-2019-42-4-425-436>). I understand this may be challenging, thus, just provide as a (future) suggestion for the authors.

R: Thanks for the constructive suggestion. We also see the challenge for this study based on a linear seismic profile to learn the lateral variation of deformations of the entire Tianshan. GPS observations and previous seismic images indicate that central Tianshan was experienced the strongest Cenozoic N–S shortening along the Tianshan Orogen. Central Tianshan is also close to the sub-horizontal underthrust Indian Slab beneath the Pamir Plateau and the West Kunlun (Fig. 1a). Thus, central Tianshan is one of the most ideal sites to learn the intracontinental deformation in response to the India-Asia collision. We introduced the uniqueness of the study region in the introduction (Lines 66–72). We appreciate your useful suggestions for the future research. We are currently processing a seismic tomography covering a large region from the Indian Craton to the Tianshan and will expect to obtain more information about the lateral variation of the Indian slab, the rotation of the Tarim Craton, and the far-field intracontinental deformation.

Response to Reviewer #3:

In this study, the authors present a 2-D Vs model and a CCP stacked receiver function image of a N-S section across the Tianshan Orogen. The seismic data were recorded by a dense linear array deployed across the orogen. While I like the careful work that the authors have done in producing the receiver function image and in obtaining the 2D S-wave model, I am not so sure whether the nice seismic results can be used here to address the scientific question on the timing of intracontinental deformation of the Tianshan Orogen, as well as whether the deformation is the response to the India-Asia collision. It is also true that you can speculate that the Tianshan orogen has a pre-weakened lithosphere before its rejuvenation in Cenozoic, but it is not a direct conclusion of your CCP image and Vs model. In conclusion, the scientific question of this study is not well phrased at the current draft, therefore I would recommend a re-submission so the authors can revise the manuscript by clearly identifying a scientific question that can be directly addressed by the seismic data here.

R: Thank you for all comments that significantly improved the manuscript. In the revised manuscript here, we rewrote the introduction to be more comprehensive and clearly focusing on the scientific topic of this study. We also reorganized some illogical statements in the discussion to avoid possible misunderstandings. It's true that seismic results have a deficiency in constraining the age of tectonic evolution. We mainly referred to the initial age of Tianshan uplift indicated by thermochronology results, and a comparison with the age estimated from seismic images can help to confirm the reasonability of the deformation model proposed in this study. We modified Fig. 1a and highlighted previous observations of the northern front of the Indian indenter that has reached the Pamir and Tarim Basin and provides an effective way to transfer the plate-convergent stress. The pre-weakened lithosphere beneath the Tianshan is also a comprehensive interpretation from low- V_S anomalies imaged in new seismic images and previous multidisciplinary studies. We hope that the revised statements will help clarify the significance of this study. Please see the following for more details. The line numbers listed in the responses are referred to the manuscript with tracking changes.

Major points:

1. You stated that "How the continental lithosphere deforms far away from plate boundary has been long debated" without saying anything about the debate and current hypotheses. It is also unclear what are the exact scientific questions on the intracontinental deformation of the Tianshan orogen that the authors want to address: the cause and timing of the Cenozoic rejuvenation, deformation style (continuous or episodic) and deformation mechanism (due to underthrust of the Tarim and Kazakh lithosphere or whole crustal/lithosphere shortening), etc. You need to provide the necessary background information on the current understanding of these questions and present a clear picture that these questions/hypotheses can be directly answered by seismic data. Same is also true for the main conclusions. Is the 10-Myr estimate of deformation history one of the main conclusions here? Has it been suggested by previous geological and/or geochemical studies? Also are there any evidence suggesting that the deformation rate is roughly constant over this 10-Myr period? Also, a pre-weakened lithosphere underneath the

orogen is a speculation, not a direct result of the seismic data.

R: We rewrote the introduction section to give a more complete picture of the regional tectonic and to concretize the scientific questions of this study (Lines 39–65). We introduced two end-member geodynamic mechanisms mainly used to interpret the driving force of the intracontinental deformation in the Tianshan (Lines 39–53). But neither of them can perfectly interpret the uplift of Tianshan, due to the inconsistency of the deformation age or the intensity of continental collision. A key to understanding the debated mechanism is the spatial and temporal configuration of the lithospheric deformation across the Tianshan. This, in turn, has significant implications for how and where the continental lithosphere responds to far-field plate collisions. However, previous studies give significant incoherent interpretations for the shallow deformation and deep structures in the Tianshan (Lines 58–65). In this study, we observe an extensively deformed lithosphere in the Tianshan, with inherited property-controlled brittle structures in the shallow crust and plastic deformation near the Moho. These features not only correlate well with the observations from geodesy and neotectonics, but also confirm that the intracontinental deformation in the Tianshan intensified synchronously with the direct contact between the Cratonic Indian slab and the Tarim Craton. Together with imaged low- V_S anomalies in the uppermost mantle beneath the Tianshan, this study has an implication for the requisite conditions of the intracontinental deformation, that is pre-weakened lithosphere and effective stress delivery.

We referred to the initial age of the Tianshan uplift from previous thermochronologic and stratigraphic studies (Fig. 1a, Lines 43–46, 160–161). We also estimated the duration of the intracontinental deformation across the Tianshan from new seismic images referring to the N-S shortening rates suggested by present-day GPS observations and Late Quaternary fault slip rates (Fig. 2e, Lines 157–161). Although seismic results have a limitation in constraining the age of tectonic evolutionary, the inferred age from seismic images is consistent with previous thermochronologic and stratigraphic results. This helps to prove the reasonability of the deformation model proposed in this study.

As shown in Figs. 2d, our results image low- V_S anomalies in the upper mantle beneath the Tianshan and suggest high temperature and weak rheology there, that is consistent with previous geophysical observations (Lines 171–178). Combining the new seismic images in this study and previous multidisciplinary studies, we suggest the presence of a hot and weak uppermost mantle beneath the Tianshan before the Cenozoic shortening.

Thank you for pointing out these confusing statements. We reorganized related statements in the discussion to avoid possible misunderstandings (Lines 155–163, 171–178).

2. *The authors show a schematic figure on balance section calculation (Figure 2f) without giving any*

details in the method. I think this observation is a key component of this study, it is better for the authors to provide more details in the method section on how to compute the N-S shortening using the identify Moho segments and crustal thickness. It seems that there are 6 Moho segments (dashed lines) shown in Figures 2c and 2d. Can you list the length of each segment, so the readers can understand where the number of ~85 km shortening comes from? Same is for the crustal volume. The two estimates are quite different (~85 km versus ~130 km), what are the likely causes of this difference?

R: Following the reviewer's suggestion, we added a new section in the method to introduce how we estimated the N–S shortening across the Tianshan (Lines 379–403). We marked the length of Moho segments and the crustal area of the 2-D cross-section to illuminate how we obtained the ~85 km and ~130 km from the balanced cross-section of the Moho length and the crustal thickening, separately (Fig. 2e). The N–S shortening estimated by the balanced cross-section of the Moho length should represent the minimum because of the limitation of seismic images in resolving small-scale thrusts. Balancing the crustal thickening also has some uncertainties caused by the variation of previous crustal thickness and the erosion of the uplifted area. We stated these limitations in the manuscript (Lines 388–390, 396–398). These estimates provide a first-order implication of the N–S shortening rate across the Tianshan, and the inferred initial age of deformation at ~10 Ma helps to confirm the proposed model.

3. I think that the authors need to provide more details on the data and parameterization either in the method section or in the supplementary document. What is the station spacing of the combined linear array? What is the grid spacing here (475 km/200 grid = 23.75 km)? It seems that the phase/group velocity data are from a previous tomography study. What is the spatial resolution of the phase/group velocity maps? What are the uncertainties in the phase/group velocity data?

R: We provided more details about the data processing in the manuscript, including the exact number of seismic stations (Lines 269–271, 405–410), the meshed grid size and resolution of the DPs (Lines 298–303), and the uncertainties of the datasets and final models (Lines 318–326). Based on the data from the MANAS network and three other stations, we construct a ~475-km long cross-section with a ~10-km station space across the central Tianshan. In total, we calculated and analyzed RFs for 43 broadband seismic stations and interpolated DPs at each station from the results of previous ambient noise tomography ($0.5^\circ \times 0.5^\circ$ grids) in the Tianshan and the Pamir Plateau. To illuminate the resolutions and uncertainties of DPs, we added a figure to show the checkerboard tests and the distributions of posterior errors in the central Tianshan where the seismic cross-section is located. Checkerboard tests indicate that group and phase velocity DPs can well resolve anomalies with a size of $0.5^\circ \times 0.5^\circ$ in the shorter periods and $1.0^\circ \times 1.0^\circ$ in the longer periods (Supplementary Fig. 3a), and posterior errors are less than 2 % for all periods (Supplementary Fig. 3b). By inverting the DPs resampled with Gaussian random noise corresponding to the posterior errors of DPs, we further estimate the errors of the final model. The

resampling tests show stable results with a standard deviation less than 1% at depths shallower than 120 km.

4. Lines 418-420: It is unclear to me why the authors can claim that *PpPs* phases have weaker sensitivities to the velocity structure than the *Ps* phases. Also, what is the V_p/V_s ratio of each 1-D V_s profile used in the computing *Ps* and *PpPs* offsets? Do you use the same V_p/V_s ratio for all the profiles?

R: We apologize for this confusion. We prepare to state that the *PpPs* phases exhibit weaker sensitivities to the V_p/V_s ratio than the *Ps* phases, which can be simply found in the κ - H slopes of *Ps* phases and *PpPs* phases (please see the figure below revised from Zhu & Kanamori [2000, doi: 10.1029/1999JB900322]). We referred to the final 1-D V_s models at each station to construct the CCP images of *PpPs* phases and *Ps* phases. Because the V_p/V_s ratios are fixed during the joint inversion, the V_p/V_s ratios at each station used in the CCP stacking are varying along the profile as we set in the initial model. They are set to the smoothed V_p/V_s ratio in the crust by averaging the recent H- κ -c stacking results²⁷ (Zhang et al., 2020) within a distance²⁷ of 50 km from each station, and set corresponding to the AK135 model in the upper mantle⁶⁹ (Kennett et al., 1995). We revised the confusion and added some more details in the manuscript (Lines 345–348, 352–358).

5. The cartoon in Figure 3 shows the Indian slab underthrusts almost horizontally to the north and collides with the Tarim cratonic mantle lithosphere. I am not sure whether this is the case at the location of the N-S profile, as the authors mention that the India slab is imaged beneath the Pamir Plateau in the west (Figure 1a). Also, the India slab shows an obvious northward dipping indicated by the intermediate seismicity (Figure 1a), so it is not underthrust horizontally. More importantly, the subducted India slab is likely an oceanic lithosphere, much weaker than a cratonic lithosphere. So, I am not sure whether this cartoon is correct here.

R: We clarified this point more clearly in the discussion (Lines 191–212). We modified Fig. 1a and highlighted previous observations of the northern front of the Indian indenter that has reached the Pamir Plateau and touched the western part of the Tarim Craton. As also mentioned by reviewer 2, this direct lithospheric contact has driven the clockwise rotation of the Tarim Craton³⁴ (Zhao et al. 2019). Plate reconstructions and global tomography suggest that before the Indian Craton arrived at the south margin of Asian Plate, >1000-km width passive continental margin, i.e., Greater India, was subducted. Due to the buoyant Indian Craton with the thick and depleted lithospheric mantle, the India-Asia convergence transited from subduction to underthrust after the break-off of the Greater India. Previous seismic images have suggested that the Cratonic Indian slab has sub-horizontally indented farther northwards beneath the western Tibetan Plateau. As the reviewer said, the northward dipping intermediate-depth seismicity beneath the Hindu Kush is suggested to be caused by the deep subduction and detachment of the Marginal Indian slab that once was the continental margin same as the Greater India⁵⁹ (Kufner et al., 2016, 2021 doi: 10.1038/s41467-021-21760-w). However, the intermediate-depth seismic zone beneath the Pamir Plateau clearly dips to the southeast-south⁶⁰ (Sippl et al., 2013 doi: 10.1002/jgrb.50128; Bloch et al., 2021) and relates to the Asian lithospheric mantle and lower crust^{33,67} (Schneider et al., 2013; Li et al., 2020), which is suggested to be a roll-back caused by the northward indentation of the buoyant Cratonic Indian slab^{56,59} (Kufner et al., 2016; Shaffer et al., 2017). Previous receiver function studies clearly confirm that the front of the Cratonic Indian slab has reached the Tarim Craton in the West Kunlun and thrust under the Pamir Plateau northward to the intermediate-depth seismic zone (Fig. 1a). Therefore, we proposed that the direct contact between the Cratonic Indian slab and the Tarim Craton enabled the compressional stresses to reach the Tianshan and caused its uplift since the Late Miocene.

Minor points:

6. Line 112: “However, the large amount of shortening (>500 km)...” It is unclear where this number comes from. The authors need provide more details.

R: We referred this to the scale of anomalies which are interpreted as the subducted Tarim Craton and Kazakh Shield in Lei & Zhao²⁰ (2007). We added the reference (Lines 139–140).

7. Figure 2a: It seems that the strain rate and GPS motion data shown in Figure 2a are not so consistent. The GPS data suggest that deformation is roughly uniformly distributed across the orogen, while the strain rates indicate a northward decreasing with spikes not necessarily collocated with faults and sutures.

R: We extracted the strain rates³⁷ (Kreemer et al., 2014) and GPS velocities⁶³ (Wang & Shen, 2020) across the cross-section from previous studies. Both the strain rate model and GPS velocity

dataset used GPS measurements in central Tianshan from Zubovich et al.²³ (2010), however, the latter includes more measurements in China and has more constraints for the southern end of the cross-section. The general decrease of GPS velocities from south to north indicates that the N–S shortening is broadly distributed throughout the Tianshan. There are some increases of the gradient of GPS velocity change close to the active thrust faults with large slip rates (>2.0 mm/yr), but these gradient variations are not obvious to distinguish from the background field of general velocity decrease. Therefore, we also extract strain rates to help to illuminate these local variations. The strain rate is generally greater than 20×10^{-9} /yr which indicates a broadly distributed deformation in the Tianshan. Local spikes reflect the activity of faults.

8. I find some inconsistency between the CCP image, and the V_s model shown in Figures 2c and 2d, respectively. For example, at -60 km, the velocity jump associated with the deeper Moho is not so clear. Also at 80-120 km, a strong negative dipping event at 10-20 km is shown in the CCP image, but you don't see a large downward velocity drop at the same depths in the V_s model? For the example station BESM shown in the Supplementary figure 4, it is better to divide the individual receiver functions recorded at this station into two groups: one from south and the other from north. As the station is located at the boundary of a Moho step, you would expect to see a significant difference in P ms between the two groups.

R: We added new synthetic tests to explore the effect of shallow structures in the CCP images (Lines 366–378, Supplementary Fig. 10). In the synthetic CCP images calculated from shallow V_s models, we can see significant negative P_s phases at depths of 10–20 km beneath the Tarim Basin and Kazakh Shield, and thus they are artifacts and definitely generated by the effect of the sedimentary layer. Our V_s model also shows normal velocities below the depth at which these artificial negative P_s phases extend in the CCP images. Therefore, the discussion is mainly focused on the structures in the Tianshan but not on the ends of the cross-section.

As the reviewer point out, we indeed observed significant azimuthal variations of RFs at some stations including the station BESM (Supplementary Fig. 41). However, these variations cannot be observed in the DPs which are more sensitive to the overall effect on absolute V_s . Thus, we used the harmonic analysis or bin-average stacking to obtain azimuthally independent RFs at each station (Lines 285–296, Supplementary Fig. 1 and 2b–d). The piercing points of the P_s phases at a depth of 60 km sample a circular area with a diameter of 0.5° centered on each station (Fig. 1b), which is similar to the horizontal grid scale of the DPs ($0.5^\circ \times 0.5^\circ$ grids) used in this study (Supplementary Fig. 3). This indicates that azimuthally independent RFs are more coordinated with the DPs, despite some details in the RFs being excluded. It's difficult to completely resolve the inconsistency between these different datasets, especially for the sharp lateral variations as the reviewer pointed out. Therefore, we detailly compared our CCP images and V_s model with previous geophysical observations along the same cross-section (Supplementary Information

Text 1, Supplementary Fig. 11) and carefully interpreted them together with previous geological results to ensure the reasonability of the proposed model.

9. Figure 2e, when you compute the crustal shortening using crustal volume, I assume you have considered the elevation. But how do you take erosion into account?

R: As mentioned in the responses above, we stated the limitations of balanced cross-section estimates in the manuscript. Variation in the initial crustal thickness and the erosion of the uplifted area will cause some uncertainties in the shortening estimates (Lines 396–398). Given that the maximum difference of the elevation across the profile is ~2 km, much less than the crustal thickness, the erosion would not significantly influence the first order estimate of shortening. Thus, we did not consider the erosion in the balanced cross-section estimates.

10. Line 383: V_p/V_s ratio of the upper mantle is 1.75. Is it too low? IASP91/AK135 has a V_p/V_s ratio of ~1.80. Will this assumption affect your inverted V_s ?

R: Following the reviewer's comment, we revised the upper mantle V_p/V_s ratio to ~1.80 according to the AK-135 model (Lines 307–310), and reconstructed the V_s models and CCP stacking images. The variations of the final models are negligible. Please see the figure below for the test of the effect of the upper mantle V_p/V_s ratio in the joint inversion. We inverted the V_s model for station KARD with different upper mantle V_p/V_s ratios, and the final models show little variations in the V_s of the upper mantle.

REVIEWER COMMENTS

Reviewer #1 (Remarks to the Author):

This is my second review of the manuscript entitled “Intracontinental deformation of the Tianshan Orogen in response to India-Asia collision” by Li et al. The revised manuscript has been greatly improved. I appreciate the efforts the authors took to address the comments from all reviewers. I have looked through all their responses and the revised manuscript and have no further comments.

Xiaotao

Reviewer #2 (Remarks to the Author):

The revised manuscript by Li W. et al. entitled 'Intracontinental deformation of the Tianshan Orogen in response to India-Asia collision' has significantly improved comparing to the first version. I am generally satisfied with the revision and just have several minor points in this round of review.

(1) The seismic cross-section of this study is close to the western end of the Tian Shan. In order for the discussion of the whole Tian Shan dynamics, it is better to specify the tectonic division of the Tian Shan at the beginning of the main text.

(2) L122-136: In my view, your seismic data can only reflect the potential lithospheric architecture of the cross-section region (or the western Tian Shan). It may be too ambitious to argue against the large-scale underthrusting of the Tarim beneath the Tian Shan. It is better to discuss this limitation and/or write more gently.

(3) L152-153: As I know, a great number of thermochronological studies of the Tian Shan have been conducted in the past half century, which provided a wider range of onset time of the Tian Shan reactivation than the data (~10 Ma) used in this paper. You may check how to deal with them.

Zhong-Hai Li

Reviewer #3 (Remarks to the Author):

In my previous review, I recommended the authors to re-present their seismic results with a clear scientific question to address. After reading the revised manuscript, I feel that the revision was insignificant, in other words, the scientific question that these seismic images can address is still unclear to me. The authors suggested that one of the mechanism responsible for the Cenozoic deformation of the Tianshan mountain range is the large-scale underthrusting of the Tarim and Kazakh lithospheres in the south and north, respectively. They stated that “In such a scenario, most of the N–S shortening would be expected to occur along the northern and southern discrete boundaries.” (lines 58-59) without providing any explanations and/or references. Intuitively, I think that underthrust-induced deformation can also be distributed across the range evenly (not necessarily limited at the southern and northern edges of the range) depending on the strength of the lithosphere beneath the range. If you can invoke a deformation mechanism with the distant collision of cratonic lithospheres between the Indica plate and the Tarim block, then you can use a similar argument to argue that underthrust of Tarim and Kazakh lithospheres can directly cause remote deformation in the central part of the range.

To me, the contributions of this study are: (1) the CCP stacking image and Vs model indicated a clear difference in crustal structure between the southern part (STAC) and the northern part (NA and IKA) of the range. The southern STAC section appears to have a pure shear type of deformation while the northern NA and IKA sections are featured by thick-skinned simple-shear deformation. (2) The seismic sections indicated that the total N-S shortening is in the range of ~85-130 km. If a shortening rate of ~12-15 mm/yr is used, then it can be inferred the shortening started at ~11-6 million years ago, which is consistent with other geologic data. I think if the authors can re-organize the manuscript with a focus on these two seismic results and their potential implications, I would recommend Nature Communications to publish this study.

On the other hand, I also have some concerns about the seismic results. First, since the dispersion data used in this study are in the period range of 6-42 s, I don't think the data have any resolutions on mantle structure below ~70 km. Therefore, it should be very cautious to make interpretation and discussion using mantle structure below 70 km. For example, the authors mentioned there is a thin high-Vs mantle lid extending to 80-90 km beneath STAC and southern NA (lines 169-170).

One important seismic structure is the three listric faults featured by an elongated low-Vs in the middle-lower crust beneath the NA and IKA sections of the range. I think the authors need show more data to

demonstrate that these fine structures are required by the data. I would suggest showing the CCP images, dispersion curves, and inverted Vs at distances of 30 km, -30 km, -90km, and -170 km (I am referring the horizontal coordinate shown in Figure 2d).

I think there are some confusion statements on displacement and deformation. For example, the authors stated that “inconsistent with the much larger shortening rate of ~5 mm/yr derived by the Global Positioning System (GPS)” (lines 103-104). I don’t think this is correct, as GPS measures velocity, not strain rate (which is the spatial derivative of the velocity field), therefore it is inappropriate to compare GPS velocity data with strain rate, like the one shown Figure 2a. The strain rate data showed a low strain rate across STAC. The slip rate of a fault measures how fast the two sides of a fault are moving relative to one another, and therefore is the difference between GPS measurements at the two sides of the fault, therefore it is different from GPS velocity.

Response to Reviewer #2:

The revised manuscript by Li W. et al. entitled 'Intracontinental deformation of the Tianshan Orogen in response to India-Asia collision' has significantly improved comparing to the first version. I am generally satisfied with the revision and just have several minor points in this round of review.

1. The seismic cross-section of this study is close to the western end of the Tian Shan. In order for the discussion of the whole Tian Shan dynamics, it is better to specify the tectonic division of the Tian Shan at the beginning of the main text.

R: Thanks for the suggestion. We've specified the tectonic division of the Tianshan in the introduction section (Lines 73-86, *note please, the line numbers listed in the responses are referred to the manuscript with tracking changes*).

2. L122-136: In my view, your seismic data can only reflect the potential lithospheric architecture of the cross-section region (or the western Tian Shan). It may be too ambitious to argue against the large-scale underthrusting of the Tarim beneath the Tian Shan. It is better to discuss this limitation and/or write more gently.

R: Many thanks for pointing out that. We've modified the manuscript accordingly (Lines 23-24, 164-165, 175-180). The seismic images constructed in this study reveal detailed crustal structures across central Tianshan and suggest that the deep crustal deformation is characterized by Moho segments and thrusts. Similar Moho thrust structures were also imaged in the eastern Tianshan (e.g., Li Y., et al., 2007⁴²; Li J., et al., 2016⁴³), indicating that the plastic deformation close to the Moho should be valid in other parts of the orogen. However, it should be noticed that the 2-D profile has limitations in confining the lateral variation of the crustal structures along the strike of the orogen. We thus pointed out the limitation and use a gentler statement as the reviewer suggested.

3. L152-153: As I know, a great number of thermochronological studies of the Tian Shan have been conducted in the past half century, which provided a wider range of onset time of the Tian Shan reactivation than the data (~10 Ma) used in this paper. You may check how to deal with them.

R: As the reviewer pointed out, some studies have recognized that Cenozoic reactivation of the Tianshan commenced at low strain at ~30-20 Ma (modest cooling) and shortening had become significant over the last ~10 Myr (rapid cooling) (simply stated in Lines 47-50, please see more details in Fig. 6 from Abdulhameed et al., 2020¹³). During the low strain stage at ~30-20 Ma, only the distant Tianshan was mildly shortened with the modest structural reactivation, whereas the southern margin of the Tianshan was loaded and subsided (moderate reheating). The rapid

shortening since ~10 Ma occurred from the Pamir Plateau to the whole Tianshan, we thus focused on this rapid cooling stage and included the onset age from previous thermochronological studies in Fig. 1a. According to the present-day GPS measurements and fault slip rates, the N–S shortening across the Tianshan derived by the seismic images can infer that the significant deformation started at ~11–6 Ma. Thus, we suggest the intracontinental deformation in the Tianshan intensified synchronously with the direct contact between the cratonic Indian slab and the Tarim Craton since the Late Miocene (~10 Ma).

Response to Reviewer #3:

In my previous review, I recommended the authors to re-present their seismic results with a clear scientific question to address. After reading the revised manuscript, I feel that the revision was insignificant, in other words, the scientific question that these seismic images can address is still unclear to me. The authors suggested that one of the mechanism responsible for the Cenozoic deformation of the Tianshan mountain range is the large-scale underthrusting of the Tarim and Kazakh lithospheres in the south and north, respectively. They stated that “In such a scenario, most of the N–S shortening would be expected to occur along the northern and southern discrete boundaries.” (lines 58-59) without providing any explanations and/or references. Intuitively, I think that underthrust-induced deformation can also be distributed across the range evenly (not necessarily limited at the southern and northern edges of the range) depending on the strength of the lithosphere beneath the range. If you can invoke a deformation mechanism with the distant collision of cratonic lithospheres between the Indica plate and the Tarim block, then you can use a similar argument to argue that underthrust of Tarim and Kazakh lithospheres can directly cause remote deformation in the central part of the range.

R: In the revised manuscript, we try to focus the scientific questions more on our new seismic images. We removed the mentioning of “large-scale underthrusting of the Tarim and Kazakh lithospheres” from the abstract (Lines 23–24) and reformulated similar statements in another place (Lines 175–180, *note please, the line numbers listed in the responses are referred to the manuscript with tracking changes*). We agree that large-scale underthrust also can cause an extensive deformation of the overriding plate. However, a significant increase of the strain rate can be expected near the ongoing underthrusting boundary where the greatest relative movement between two blocks is located, such as the case in the India-Asian collision zone (Lines 65–72). In the Himalayas, where the Indian slab is underthrusting beneath the Tibetan Plateau, the strain rate increases to $\sim 100 \times 10^{-9}/\text{yr}$ along the belt and is much larger than that inside the Tibetan Plateau (Fig. 1a). High strain rates also concentrate along the northern front of the Pamir Plateau, where the Asian Plate is subducting below it (Fig. 1a). However, the distribution of high strain rates across the central Tianshan apparently indicates a different type of deformation.

To me, the contributions of this study are: (1) the CCP stacking image and Vs model indicated a clear difference in crustal structure between the southern part (STAC) and the northern part (NA and IKA) of the range. The southern STAC section appears to have a pure shear type of deformation while the northern NA and IKA sections are featured by thick-skinned simple-shear deformation. (2) The seismic sections indicated that the total N-S shortening is in the range of ~85-130 km. If a shorting rate of ~12-15 mm/yr is used, then it can be inferred the shortening started at ~11-6 million years ago, which is consistent with other geologic data. I think if the authors can re-organize the manuscript with a focus on these two seismic results and their potential implications, I would recommend Nature Communications to publish this study.

R: The reviewer summarized our main findings appropriately, which we appreciate very much. We reorganized the abstract and discussion to emphasize the two points (Lines 20–33, 177–183, 185–264). In our manuscript, the two points are presented and discussed in detail in the two sections following the introduction and results. Point (1) is extensively discussed in section “Inherited properties control shallow crustal shortening” and point (2) in section “Moho deformation responses to deep crustal shortening”.

In the last section, “Rejuvenation of the Tianshan Orogen”, we extended the discussion to infer how the deformations distributed within the Tianshan were intensified in the last 10 million years and by which tectonic triggers. We argue that the rejuvenation of the Tianshan was triggered by the direct contact of the underthrusting Indian slab with the Tarim Craton’s mantle lithosphere beneath the Pamir Plateau and West Kunlun. This extension is closely related to our seismic images, and we think it will not only attract a broader readership but is also important and timely. It has long been an issue of debate how far north the Indian mantle lithosphere is underthrusting beneath Tibet. Only very recently, seismic studies have provided clear evidence that the Indian slab front has reached Pamir and west Tibet and contacts the Tarim Craton (summarized in Lines 210-216, Fig. 1a). At the AGU Fall Meeting 2021, Simon Klemperer et al. presented solid evidence from mantle earthquakes for the northern front of the Indian slab beneath the border of western Tibetan Plateau and Tarim Craton (<https://agu.confex.com/agu/fm21/meetingapp.cgi/Paper/925787>). The Indian slab front is along the border between Tarim and West Tibet–Pamir and follows exactly the Pamir-Hindu Kush intermediate-depth seismic zone, recently accurately defined by Bloch et al.⁶⁰ (GRL, 2021). This direct interaction of the Indian indenter with the rigid Tarim Craton has profound consequences on the Cenozoic evolution of the Tianshan orogenic belt, but was so far not considered by most of the previous studies of Tianshan.

On the other hand, I also have some concerns about the seismic results. First, since the dispersion data

used in this study are in the period range of 6-42 s, I don't think the data have any resolutions on mantle structure below ~70 km. Therefore, it should be very cautious to make interpretation and discussion using mantle structure below 70 km. For example, the authors mentioned there is a thin high-Vs mantle lid extending to 80-90 km beneath STAC and southern NA (lines 169-170).

R: The reviewer is right with the reduced sensitivity of surface waves at the mantle depths. The uppermost mantle is mainly resolved by receiver functions. In the revised manuscript we are more cautious with the mantle structures and reduced the weights of the corresponding discussions. We also did some tests to assess the resolution for the sub-Moho depths. At first, we performed the robustness tests of the mantle lid structure imaged beneath the STAC and the NA, which can be recognized in the 1-D V_S models beneath stations GOLB and KARD. We modified the 1-D V_S models of these stations by excluding high- V_S mantle lid at depths of ~50–80 km and calculated synthetic RFs and DPs. Both synthetic DPs from the final and the modified 1-D V_S models are similar to the observed ones, but the synthetic RFs from the modified models fail to fit the negative P_s phases at ~6–8 s. These features indicate that the mantle lid structure is reliable although DPs used in this study have a limited resolution of the mantle structure (Lines 382–388, Supplementary Figs. 5d and 5e). Moreover, the interpretation of high temperature and weak rheology in the mantle refers to previous geophysical observations (Lines 236–238) and is suggested by the scattered small-volume basalts and high heat flux values inferred from xenoliths (Lines 239–241). The consistency between our images and previous studies also gives confidence in the reliability of the result (Lines 244–245).

One important seismic structure is the three listric faults featured by an elongated low-Vs in the middle-lower crust beneath the NA and IKA sections of the range. I think the authors need show more data to demonstrate that these fine structures are required by the data. I would suggest showing the CCP images, dispersion curves, and inverted Vs at distances of 30 km, -30 km, -90km, and -170 km (I am referring the horizontal coordinate shown in Figure 2d).

R: We provided more details about the data fitting and robustness tests of the inversion at stations GOLB, KARD, BESM, and SOUR (Lines 367–388, Supplementary Figs. 4 and 5). These stations are close to the locations suggested by the reviewer. Three elongate low- V_S anomalies beneath the NA and IKA in the 2-D V_S model (Fig. 2d) can be easily recognized at depths of ~20–50 km in the 1-D V_S models beneath stations KARD, BESM, and SOUR, respectively. We modified the 1-D V_S models of stations KARD, BESM, and SOUR, by excluding the crustal low- V_S anomalies, and calculated synthetic RFs and DPs to test the reliability of the low- V_S anomalies. We find significant mismatches between the observed and calculated RFs and DPs (Supplementary Figs. 5a–c), which indicate that the low- V_S anomalies are reliable features

derived from the data. We also performed the robustness tests of the mantle lid structure imaged beneath the STAC and the NA and confirmed the reliability of the mantle lid structure (Supplementary Figs. 5d and 5e).

I think there are some confusion statements on displacement and deformation. For example, the authors stated that “inconsistent with the much larger shortening rate of ~5 mm/yr derived by the Global Positioning System (GPS)” (lines 103-104). I don’t think this is correct, as GPS measures velocity, not strain rate (which is the spatial derivative of the velocity field), therefore it is inappropriate to compare GPS velocity data with strain rate, like the one shown Figure 2a. The strain rate data showed a low strain rate across STAC. The slip rate of a fault measures how fast the two sides of a fault are moving relative to one another, and therefore is the difference between GPS measurements at the two sides of the fault, therefore it is different from GPS velocity.

R: We modified the statements accordingly (Lines 123–128). In the previous manuscript, we derived the N-S shortening rate of ~5mm/yr across the STAC from the difference of the GPS velocities at its southern (NTF, ~15mm/yr) and northern (AIF, ~10mm/yr) boundaries (Fig. 2a). To avoid the potential confusion of the shortening rate indicated in this point and the strain rate which is the spatial derivative of the GPS velocity field, we deleted the related statement in the revised manuscript.

REVIEWERS' COMMENTS

Reviewer #3 (Remarks to the Author):

I think the authors have done their best in responding my comments. Thus, I am fine with publishing the manuscript in its current form.